# Single-cell transcriptomic analysis of the tumor ecosystems underlying initiation and progression of papillary thyroid carcinoma

Weilin Pu[1,2,12], Xiao Shi[1,3,12], Pengcheng Yu[1,3,12], Meiying Zhang[4,12], Zhiyan Liu [5], Licheng Tan[1,3], Peizhen Han[1,3], Yu Wang[1,3], Dongmei Ji [3,6,7], Hualei Gan[3,8], Wenjun Wei[1,3], Zhongwu Lu[1,3], Ning Qu[1,3], Jiaqian Hu[1,3], Xiaohua Hu[2], Zaili Luo[9], Huajun Li[10], Qinghai Ji[1,3], Jiucun Wang [2,11], Xiaoming Zhang [4✉] & Yu-Long Wang [1,3✉]

The tumor ecosystem of papillary thyroid carcinoma (PTC) is poorly characterized. Using single-cell RNA sequencing, we profile transcriptomes of 158,577 cells from 11 patients' paratumors, localized/advanced tumors, initially-treated/recurrent lymph nodes and radio-active iodine (RAI)-refractory distant metastases, covering comprehensive clinical courses of PTC. Our data identifies a "cancer-primed" premalignant thyrocyte population with normal morphology but altered transcriptomes. Along the developmental trajectory, we also discover three phenotypes of malignant thyrocytes (follicular-like, partial-epithelial-mesenchymal-transition-like, dedifferentiation-like), whose composition shapes bulk molecular subtypes, tumor characteristics and RAI responses. Furthermore, we uncover a distinct *BRAF*-like-B subtype with predominant dedifferentiation-like thyrocytes, enriched cancer-associated fibroblasts, worse prognosis and promising prospect of immunotherapy. Moreover, potential vascular-immune crosstalk in PTC provides theoretical basis for combined anti-angiogenic and immunotherapy. Together, our findings provide insight into the PTC ecosystem that suggests potential prognostic and therapeutic implications.

[1] Department of Head and Neck Surgery, Fudan University Shanghai Cancer Center, Shanghai 200032, China. [2] State Key Laboratory of Genetic Engineering, Collaborative Innovation Center for Genetics and Development, School of Life Sciences, Fudan University, Shanghai 200438, China. [3] Department of Oncology, Shanghai Medical College, Fudan University, Shanghai 200032, China. [4] The Center for Microbes, Development and Health, Key Laboratory of Molecular Virology & Immunology, Institut Pasteur of Shanghai, Chinese Academy of Sciences, Shanghai 200031, China. [5] Department of Pathology, Shanghai Jiao Tong University Affiliated Sixth People's Hospital, Shanghai 200233, China. [6] Department of Medical Oncology, Fudan University Shanghai Cancer Center, Shanghai 200032, China. [7] Phase I Clinical Trial Center, Fudan University Shanghai Cancer Center, Shanghai 200032, China. [8] Department of Pathology, Fudan University Shanghai Cancer Center, Shanghai 200032, China. [9] Brain Tumor Center, Division of Experimental Hematology and Cancer Biology, Cincinnati Children's Hospital Medical Center, Cincinnati, OH 45229, USA. [10] Department of Clinical Research & Development, Jiangsu Hengrui Pharmaceuticals Co., Ltd., Shanghai 201210, China. [11] Human Phenome Institute, Fudan University, Shanghai 200438, China. [12] These authors contributed equally: Weilin Pu, Xiao Shi, Pengcheng Yu, Meiying Zhang. ✉email: xmzhang@ips.ac.cn; yulongwang@fudan.edu.cn

The incidence of thyroid cancer has increased by 3% annually in the United States over the last four decades, driven largely by the rise in papillary thyroid cancer (PTC)[1]. Although most PTCs present an indolent clinical course, some of them have developed to locoregional or even distant metastatic disease at diagnosis. In the recurrent or metastatic settings, combination of surgery, radioactive iodine (RAI) ablation and thyroid stimulating hormone (TSH) suppression can still achieve a favorable prognosis for most cases, while a fraction of patients would ultimately progress into RAI-refractory (RAIR) status or even succumb to this disease, who may be potential candidates for alternative treatments including molecular targeted inhibitors and immunotherapies[2,3]. The evolving trends of progressive PTCs have challenged the clinical practice and promoted researchers to further decipher their biological architectures.

Bulk sequencing of PTC have advanced our understanding of its genetic characteristics[2,4,5]. For instance, detection of $BRAF^{V600E}$ and $TERT$ promoter mutations can help distinguish malignant thyroid nodules and identify patients with dedifferentiation potential[5,6]. However, the biological underpinnings of PTC evolution from early to advanced stage, or from RAI-avid to RAIR state remain unclarified. Although bulk sequencing can delineate the genetic landscape of the whole tumor entity, it inevitably averages the expression profiles of diverse cells and masks the critical differences between tumor components. This highlights a critical need to elucidate the compositions, properties and underlying mechanisms in the complex tumor microenvironment (TME). Single-cell RNA sequencing (scRNA-seq), which enables us to quantify features of individual cells, is a powerful tool for the investigation of the cellular components and their interactions in the TME. Currently, scRNA-seq has been widely applied in a broad spectrum of cancers[7]. However, improved characterization of PTCs at single-cell resolution is still lacking.

In this work, we perform scRNA-seq to analyze 158,577 cells from 11 PTC patients' paratumors, localized or advanced tumors, initially-treated or recurrent lymph nodes (LNs), and RAIR distant metastases, covering comprehensive clinical courses of this disease, filling the current blank of single-cell profiling of human PTC. Using this unique resource, we analyze cell lineages, transcriptional states, developmental trajectories and cell-cell crosstalk in PTCs, thereby shedding light on the tumor ecosystems underlying PTC initiation and progression.

## Results

**A single-cell expression atlas of papillary thyroid cancer ecosystems**. To comprehensively resolve the tumor ecosystem heterogeneity during PTC initiation and progression, we used scRNA-seq (10X Genomics) to profile tumor and stromal cells of 23 fresh samples from 11 patients (Fig. 1a), including six paratumor tissues, seven primary tumors, eight involved LNs, and two RAIR subcutaneous loci belonging to distant metastases in PTC. In addition to classical PTCs, three patients were diagnosed with follicular variant (FV, Case 5 and 7) or tall-cell variant (TCV, Case 6) after careful pathological review. Detailed clinicopathological information is provided in Supplementary Data 1. Moreover, the genomic mutations for these patients were also assessed by whole-exome sequencing (WES) and Sanger sequencing, and the status of key PTC-related mutations (BRAF, RAS, TERT promoter) was provided (Supplementary Data 2; Supplementary Table 1). Except for Case 4, 6, and 7 with only recurrent loci, the remaining eight patients had paired samples collected for scRNA-seq analysis. For example, for case 11, the involved LNs and subcutaneous distant loci were identified by computed tomography (CT) scan, clinical examination and pathological review to assure the correct tissues for analysis (Supplementary Fig. 1a, b). By this means, our cohort covered a relatively comprehensive collection of tissues mirroring tumor progression process, including paratumors, primary lesions, nodal metastases and RAIR distant metastases.

A total of 158,577 single cells with a median of 1,215 expressed genes passed the stringent quality filtering and were incorporated in further analysis (Fig. 1b; Supplementary Table 2). After integrating the transcriptional data from all acquired cells, we primarily applied low-resolution t-distributed stochastic neighbor embedding (t-SNE) clustering and identified six main cell populations, which were labeled as T/natural killer (NK) cells (CD3D, CD3E, CD3G, CD247), B cells (CD79A, CD79B, IGHM, IGHD), thyrocytes (TG, EPCAM, KRT18, KRT19), myeloid cells (LYZ, S100A8, S100A9, CD14), fibroblasts (COL1A1, COL1A2, COL3A1, ACTA2) and endothelial cells (PECAM1, CD34, CDH5, VWF) (Fig. 1c; Supplementary Fig. 1c). Each of these populations was captured from different tissue types of different patients (Supplementary Fig. 1d, e). In addition, all these cell types were further validated using another scRNA-seq study of PTC (Supplementary Fig. 1f, g)[8].

Subsequently, we aimed to depict a more detailed immune landscape through a high-resolution t-SNE analysis. The T, NK, B, and myeloid cell lineages were demarcated into 22 finer subclassifications based on their patterns of differentially expressed genes (DEGs) (Fig. 1d). Specifically, T cells were dichotomized according to the expression level of CD4 and CD8, which were further divided into 7 clusters for CD4+ cells and 4 clusters for CD8+ cells, while the NK, B, and myeloid cells were divided into 2, 3, and 6 subsets, respectively (Fig. 1d). Each cluster was assigned with a putative identity demonstrating its potential functional capabilities and demonstrated different tissue enrichment preferences, as quantified by the ratio of observed to expected cell numbers in each cluster (Ro/e) in previous reports[9,10] (Fig. 1e; Supplementary Table 3).

For example, CD4-c6 and CD8-c4 clusters expressed upregulated levels of FOXP3 and PDCD1 (encoding programmed cell death protein-1, PD-1), corresponding to regulatory CD4+ T cells (Treg) and exhausted CD8+ T cells (Tex), respectively[9], both of which negatively regulated antitumor response (Supplementary Table 3). In addition, the CD8-c3 cluster was characterized by upregulated cytotoxic marker gene GNLY (encoding granulysin), which might act as effector T cells[11] (Supplementary Table 3). Compared with paratumor samples, Tregs, Texs, and effector T cells were all enriched in tumor tissues, suggesting the coexistence of host immune response and tumor immune escape in the PTC milieu (Fig. 1e). Furthermore, the ISG15-expressing CD4+ T cells were significantly enriched in paratumors. Previous reports suggested that ISG15 might participate in natural killer (NK) cell proliferation, dendritic cell maturation, or other innate immune responses[12–14]. However, the definite role of this subcluster in PTC needs to be further examined.

However, the cytotoxic CD8-c3-GNLY cells were not enriched in the subcutaneous loci, replaced by the abundance of Tregs and Texs and another two immune cell clusters, CD8-c1 and Mφ-c3 (Fig. 1e). The CD8-c1 cluster had high expression of GZMK, which was reported to be an intermediate state between effector and exhausted T cells[15], while the Mφ-c3 cluster exhibited high levels of CCL18 that are upregulated in the pro-tumor M2 macrophages[16,17]. Meanwhile, the Mφ-c1-RGS1 cluster with phagocytic potential appeared to be sparse in this region (Fig. 1e; Supplementary Table 3). These results indicated that formation and development of subcutaneous metastases in PTC may require a more immunosuppressive TME than that of primary tumors and lymphadenopathies.

**Single-cell transcriptional profiles of thyrocytes reveal a tissue origin-related pattern**. To explore the single-cell transcriptional

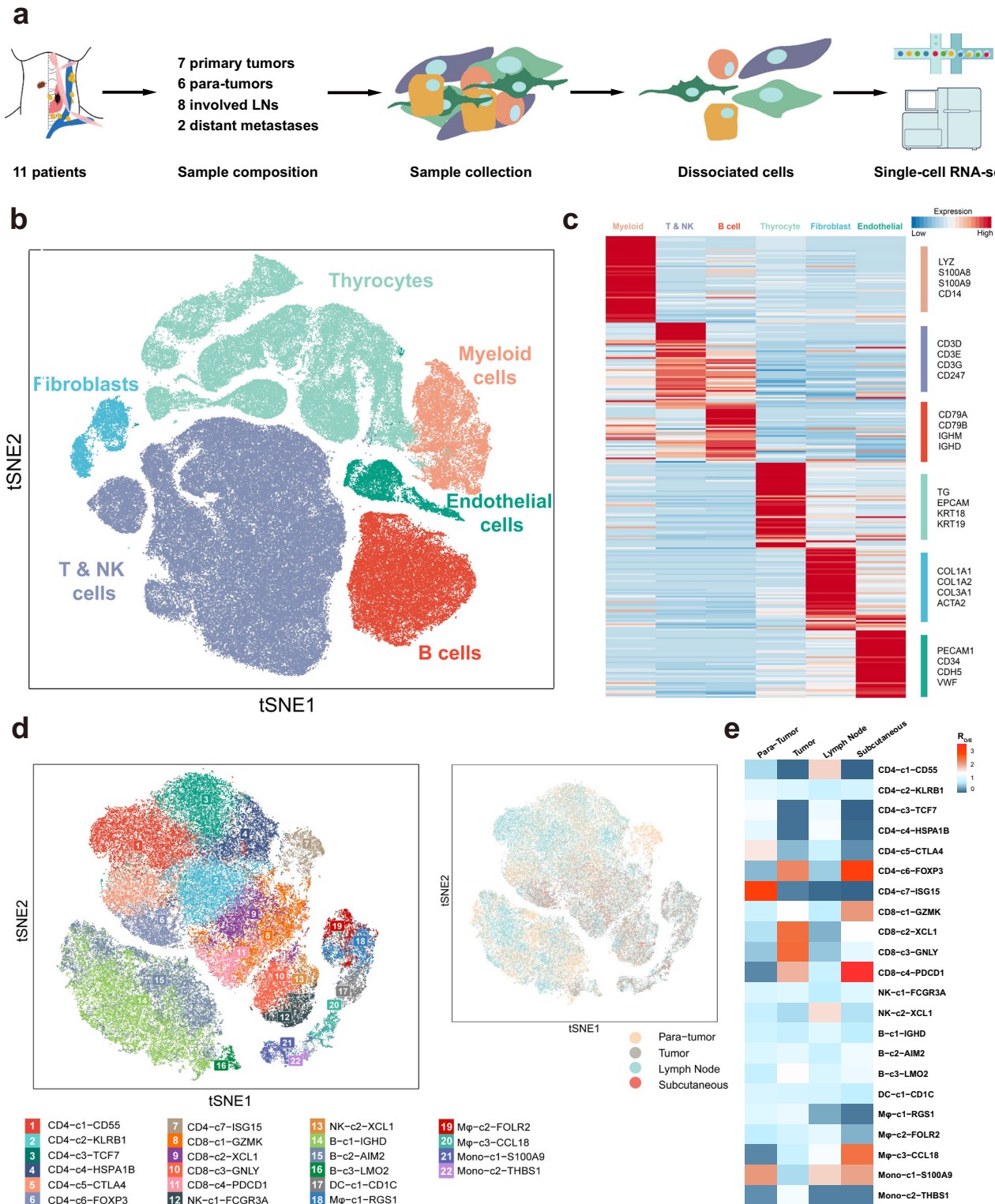

**Fig. 1 Expression profiling of 158,577 single cells in PTCs. a** Workflow of sample composition, processing and bioinformatic analyses for 23 samples in the present study. **b** t-SNE plot of all high-quality cells profiled in the present study colored by major cell lineage. **c**, Heatmap of the canonical and curated marker genes for major cell lineages. **d** t-SNE projection showing the landscape of immune cells, colored by cluster (left) and tissue (right). **e** Tissue preference for each immune cell subcluster estimated by *Ro/e*. Source data are provided in the Source Data file.

heterogeneity of thyrocytes, we used Uniform Manifold Approximation and Projection (UMAP), a nonlinear dimensionality-reduction technique[18], to characterize them into 9 different clusters along the epithelial cell lineages (Fig. 2a). As visualized by cluster distributions in the UMAP plot, expression programs of thyrocytes revealed substantial heterogeneities, but appeared to be closely related with their sample origins (Fig. 2b). Moreover, we found that c03, c05, c06, and c09 are mainly defined by cells from single sample. Therefore, we then performed the pathway enrichment analysis using the DEGs for each cluster and identified distinct up-

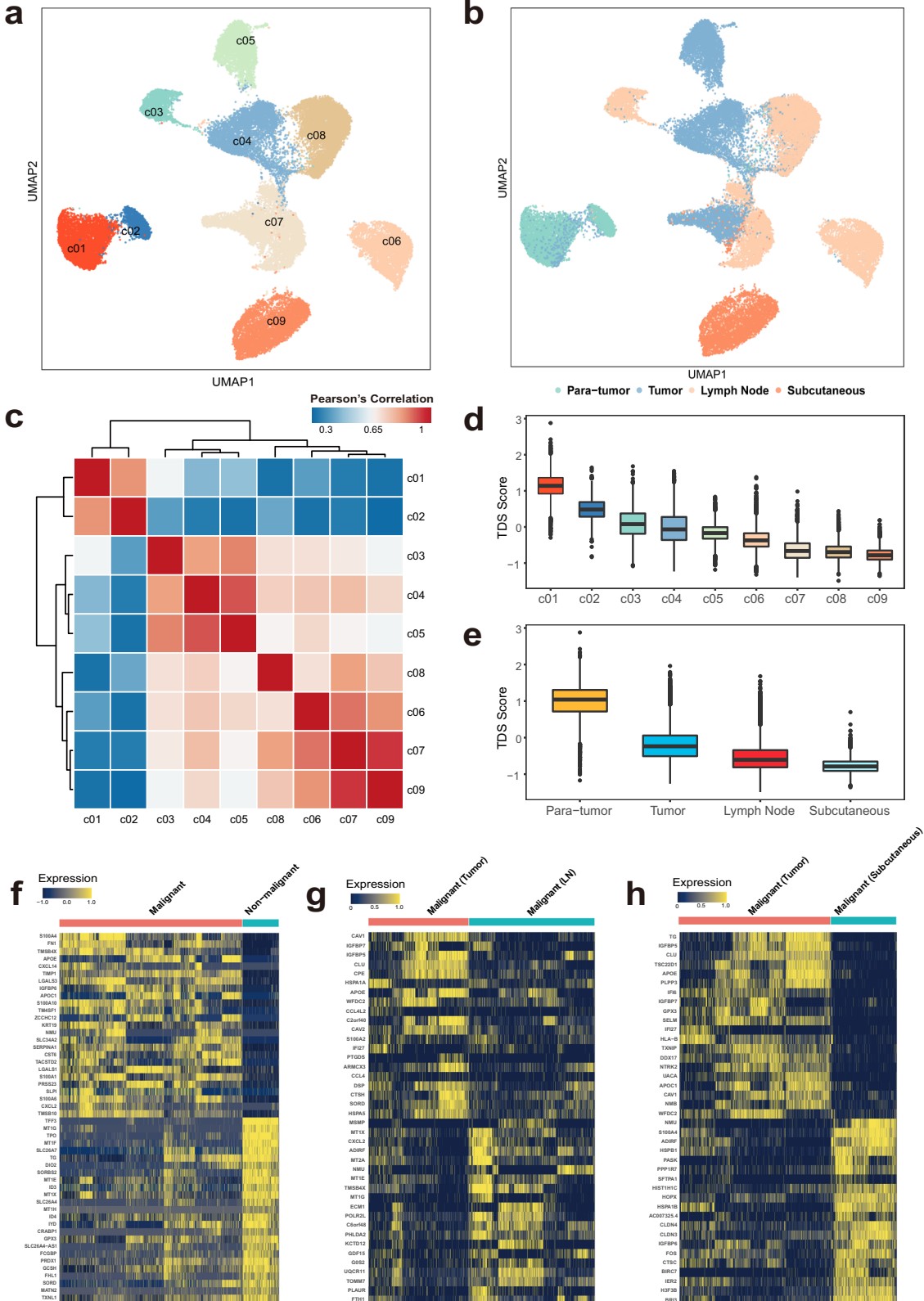

**Fig. 2 Identification of malignant thyrocytes and their transcriptional heterogeneity between different tissue types.** UMAP projection of 36,265 thyrocytes colored by (**a**) cluster and (**b**) tissue. **c** Heatmap of pair-wise Spearman's correlations among c01-c09 thyrocyte clusters. Boxplots showing TDS scores of (**d**) each thyrocyte cluster and (**e**) tissue type. The number of cells in each group is shown in Supplementary Table 4. The middle lines of the boxplots show the median (central line), the lower and upper hinges show the 25–75% interquartile range (IQR), and the whiskers extend from the hinge to the farthest data point within a maximum of 1.5x IQR. Heatmaps of the top DEGs between (**f**) malignant versus non-malignant thyrocytes, (**g**) tumor-derived versus LN-derived malignant thyrocytes, and (**h**) tumor-derived versus subcutaneous loci-derived malignant thyrocytes. Source data are provided in the Source data file.

regulated and down-regulated pathways for each cluster, suggesting that their differences were attributed to biological diversities cause by inter-tumor heterogeneities rather than batch effects (Supplementary Table 4; Supplementary Data 3). Among the 9 clusters, c01 and c02 distributed away from other populations, and both populations mainly derived from paratumors. In turn, the vast majority of paratumor cells fell into c01/c02 clusters, while c03-c09 cells were almost entirely composed of thyrocytes from primary tumors, LNs and subcutaneous metastatic samples (Fig. 2b, Supplementary Fig. 2a, b; Supplementary Table 4), suggesting that c01/c02 and c03-c09 populations might represent non-malignant and malignant thyrocytes, respectively.

To further validate this presumption, we clarified their differences by three complementary approaches. First, we calculated the average expression programs of all clusters, in which c01 and c02 were highly correlated (Pearson's $R = 0.82$), while the remaining seven clusters revealed a closer connection (Pearson's $R > 0.62$), suggesting the diverse transcriptional profiles of c01/c02 with other populations (Fig. 2c). Second, we calculated each thyrocyte's thyroid differentiation score (TDS), which is a widely used algorithm to evaluate the differentiation status of PTC[4]. Consistently, c01/c02 thyrocytes had higher TDS values and upregulated thyroid differentiation-related genes such as TFF3, TPO, TG, and DIO2, while c03-c09 clusters had lower TDS scores and increased expression of PTC-related genes, such as S100A4, FN1, IGFBP6, and KRT19 (Fig. 2d; Supplementary Fig. 2c, d). Third, we constructed a machine learning classifier based on bulk transcriptional profiles from PTC cases in The Cancer Genome Atlas (TCGA) dataset ("Methods"), which was successfully validated with 97% sensitivity and 96% specificity in an additional bulk RNA-seq cohort from Yoo et al.'s study[19] (available in EBI European Nucleotide Archive database with accession number PRJEB11591) (Supplementary Table 5). Subsequently, this classifier was applied in our scRNA-seq profiles and exhibited a well distinguishable ability, with 95%, 97% and > 98% of c01, c02, and c03–09 cells in accordance with their putative non-malignant or malignant identities, respectively (Supplementary Table 6). Together, these data verified the accuracy of our approach that confidently distinguished malignant (c03-c09) and non-malignant (c01, c02) thyrocytes in PTCs.

For malignant compartments, thyrocytes from metastatic lesions generally had lower degrees of differentiation than those from primary tumors (Fig. 2e). Therefore, in addition to the overall differences between malignant and non-malignant thyrocytes (Fig. 2f), transcriptional heterogeneities within the malignant component are also worth further investigation. Compared with malignant cells from primary tumors, their nodal metastatic counterparts were characterized by upregulation of genes (MT1X, MT2A, MT1E, MT1G) in the metallothionein family (Fig. 2g). On the other hand, consistent with the low TDS score, post-RAI subcutaneous metastatic thyrocytes lacked TG while preferentially expressed a set of genes associated with epithelial-mesenchymal transition (EMT) (CLDN3, CLDN4), cell cycle (S100A4, HSPB1) and stress responses (FOS, IER2) (Fig. 2h). In addition, we identified 22 genes to be significantly and positively associated with TDS score (Pearson's $R > 0.5$, $P < 0.05$) at single-cell resolution (Supplementary Table 7). Among them, eight genes (MT1F, SORBS2, MT1G, SORD, SLC26A4-AS1, PRDX1, FCGBP, MATN2) have not been reported to be involved in thyroid differentiation. The functions and roles of these genes in PTC tumorigenesis deserve further exploration.

Among the most dysregulated DEGs ($\log_2$ fold-change > 2, FDR < 0.05) between different origins of thyrocytes, we found TMSB4X, which has not been reported in PTC, was significantly upregulated in malignant cells than in non-malignant cells (Fig.2f; Supplementary Table 8). Furthermore, we also observed an increasing pattern of TMSB4X expression from primary cancer cells to LN-metastatic cells (Fig.2g). Consistent with our scRNA-seq findings, both the TCGA and PRJEB11591 cohorts validated the incremental trend of TMSB4X expression from paratumor to N0-stage tumors to N1-stage tumors in bulk profiles (Supplementary Fig. 3a, b). At the protein level, immunohistochemistry (IHC) staining also revealed an evidently higher expression of TMSB4X on tumor cells compared with their adjacent normal thyrocytes in two additional cases (a classical PTC and an FV-PTC, Supplementary Fig. 3c, d). Taken together, these data highlighted TMSB4X as a suggestive biomarker that potentially involves in PTC initiation and progression.

**Identification of premalignant thyrocyte population harboring cancer-primed properties**. As shown in Fig. 2d, the inconsistency in TDS scores between c01 and c02 cells suggested heterogeneous expression programs within non-malignant compartments. We then sought to explore the differences in single-cell transcriptomes between the two populations. The t-SNE analysis indicated that c01 and c02 cells represented two distinct states of non-malignant thyrocytes (Fig. 3a), in which the c02 cluster was enriched in pathways associated with cell proliferation (OXIDATIVE_PHOSPHORYLATION, MYC_TARGETS_V1) and stress response (DNA_REPAIR, HYPOXIA) according to the Kyoto Encyclopedia of Genes and Genomes (KEGG) analysis (Supplementary Fig. 4a). Moreover, trajectory analysis further suggested the potential transitions between c01 and c02 population (Fig. 3b). Specifically, from c01 to c02 to malignant (c03–09) clusters, we observed a descending trend of classical thyroid epithelial markers including TG, TPO and IYD and an ascending trend of TMSB4X that increased along with PTC initiation and progression (Fig. 3c, Supplementary Fig. 4b). Therefore, in terms of transcriptional profiles, the c02 cluster might not be completely normal thyrocytes, but instead represented a premalignant state.

To evaluate histological features of the premalignant c02 cluster, we then reviewed the hematoxylin-eosin (H&E)-stained slides of Case 5 whose paratumor tissues (P5) donated the majority of thyrocytes in this cluster. From pathological sections of the P5 sample, we confirmed the non-malignant nature of c02 cells by their histologically normal follicular architectures, indicating the aberrant expression programs have yet to bring about evident changes in cellular morphology (white arrow, Fig. 3d). However, although preoperative examinations lacked positive findings in the region where P5 was obtained, we observed multifocal occult tumor foci (yellow arrow) embedded in these c02 thyrocytes (white arrow) in H&E-stained sections (Fig. 3d).

Then we evaluated the differences between P5 and T5 (primary tumor) malignant cells. Compared with their T5 counterparts, P5 occult cancer cells not only had diverse morphologic features and higher TDS scores, but also occupied evolutionary positions closer to the P5 premalignant cells (Fig. 3d, e, f), suggesting they were more likely to originate from their neighboring premalignant cells rather than from the primary lesion through intraglandular metastases. In other words, these data supported the cancer-primed nature of these outwardly normal but transcriptionally altered premalignant cells (c02), which presumably provide both seeds and soil for the eruption of malignant growths.

The most dysregulated genes ($|\log_2$ fold-change$| > 1$, FDR < 0.05) between normal (c01) and premalignant (c02) thyrocytes reflected the underlying early-onset transcriptional changes during PTC tumorigenesis (Fig. 3g; Supplementary Fig. 4b), in which PKHD1L1 was significantly downregulated in premalignant and malignant thyrocytes (Supplementary Fig. 4c;

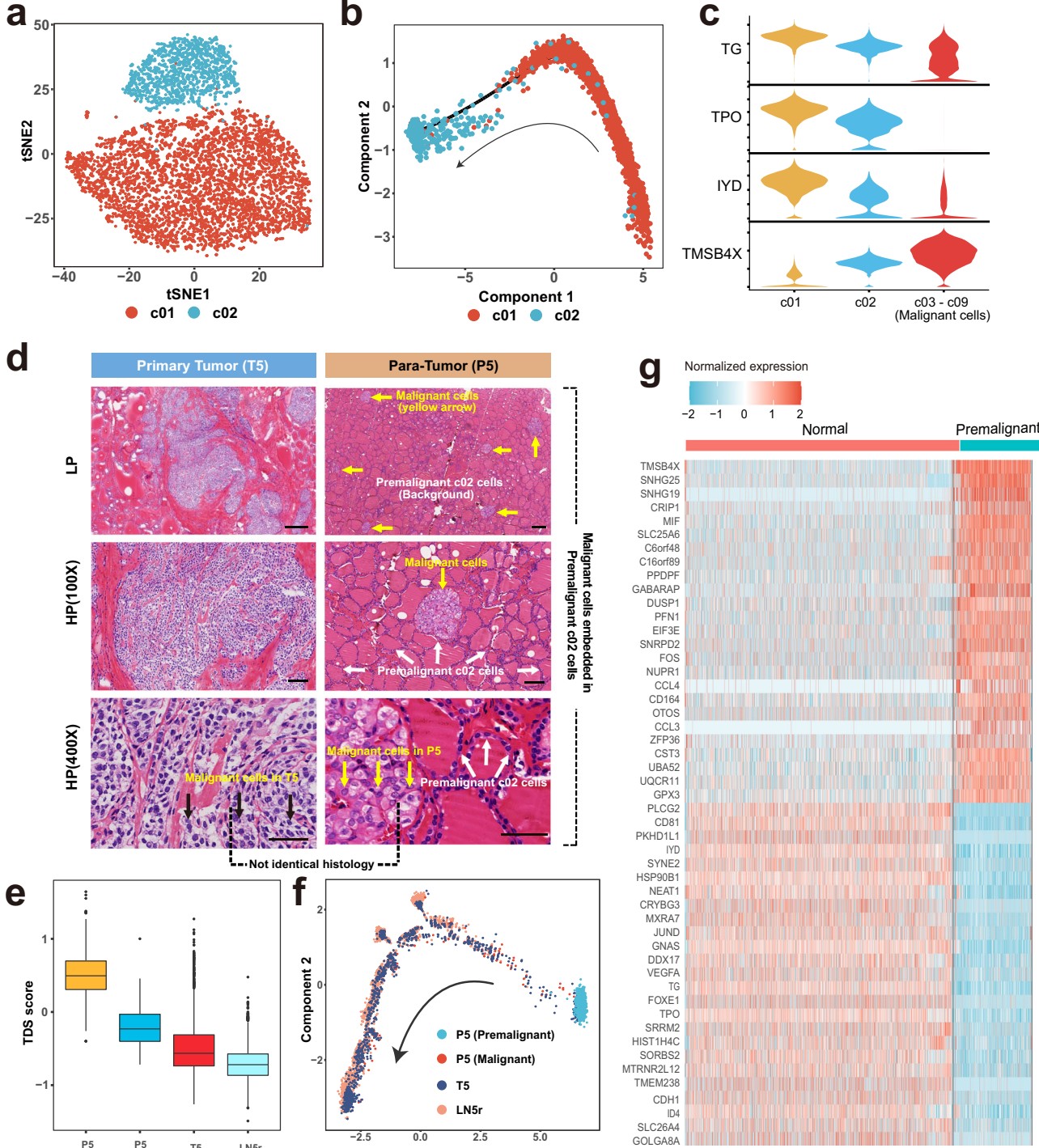

**Fig. 3 Identification and transcriptional characterization of premalignant thyrocytes. a** t-SNE projection of the two clusters of non-malignant thyrocytes (c01 and c02). **b** Potential trajectory of c01 and c02 cells inferred by Monocle2. The arrow shows the putative developmental direction from c01 to c02 cluster. **c** Violin plots showing several DEGs between c01, c02, and c03-c09 clusters. Expression level of *TG*, *TPO*, and *IYD* decreases while the level of *TMSB4X* increases continuously from c01 to c02 and to c03-c09 thyrocytes. **d** Representative H&E staining of primary tumor (T5) and para-tumor (P5) of Case 5 at magnifications of × 20, × 100, and × 400. Three independent experiments were performed and generated similar results. Scale bar (upper) = 300 μm; Scale bar (middle) = 100 μm; Scale bar (bottom) = 50 μm. **e** Boxplot showing TDS scores of thyrocytes from the premalignant paratumor (*n* = 1036), malignant paratumor (*n* = 118), tumor (*n* = 1337) and lymph node metastasis (*n* = 4986) of Case 5. The middle lines of the boxplots show the median (central line), the lower and upper hinges show the 25–75% interquartile range (IQR), and the whiskers extend from the hinge to the farthest data point within a maximum of 1.5 x IQR. **f** Trajectory analysis of four thyrocyte populations from Case 5 (P5 premalignant cells, P5/T5/LN5r malignant cells). The arrow shows potential evolutionary direction from P5 premalignant thyrocytes to P5/T5/LN5r malignant thyrocytes. **g** Heatmap of normalized expression of top upregulated and downregulated genes between normal (c01) and premalignant thyrocytes (c02). Source data are provided in the Source data file.

Supplementary Table 9). In bulk profiles, we observed a decreased *PKHD1L1* expression in tumor samples compared with that in normal thyroid tissues (TCGA cohort, Supplementary Fig. 4d). Meanwhile, we also found a downward *PKHD1L1* expression with the decline of follicular patterns across different histologic subtypes (PRJEB11591 cohort, Supplementary Fig. 4e). Furthermore, in the TCGA dataset, downregulation of *PKHD1L1* not only predicted a significantly compromised disease-free survival (DFS, *P* = 0.0063) in thyroid cancer (Supplementary Fig. 4f), but also predicted a poorer overall survival (OS) in lung adenocarcinoma (*P* = 0.0013) and melanoma (*P* = 1.2e−6) (Supplementary Fig. 4g, h), suggesting that *PKHD1L1* might function as a tumor suppressor gene in multiple solid tumors.

**Developmental trajectory defines distinct states of malignant thyrocytes associated with tumor characteristics and response to RAI treatment.** The collection of four types of lesions from different clinical course of PTCs gave us an opportunity to dissect the evolutionary dynamics of thyroid epithelial lineages. To mirror this process, we applied Monocle to perform trajectory inference of all acquired thyrocytes. We found that the trajectory-estimated pseudotime fit well with the variation trend of TDS, *BRAF* and *RAS* scores (Fig. 4a; Supplementary Fig. 5a, b), indicating a good correlation between pseudotime progression and an increased malignant degree. Subsequently, our trajectory analysis yielded three developmental hierarchies (State 1–3) where the normal c01 and premalignant c02 clusters located at the top-right corner, suggesting a clear starting point of cell evolution on this map (Fig. 4b; Supplementary Table 10). After confirmation of this starting point, developmental routes were clearly determined beginning with the normal-cell-initiated State 1 and then bifurcating into either State 2 or State 3 branches with metastasis-rich endpoints (Fig. 4b).

For these three states, (1) almost all paratumor thyrocytes with normal follicular epithelial features concentrated in State 1, which appeared to be an indolent state reflected by malignant scores (Fig. 4c; Supplementary Fig. 5c). Meanwhile, single-cell regulatory network inference and clustering (SCENIC) analysis also predicted an increased expression of *SOX9*, a key transcription factor (TF) in branching folliculogenesis of normal thyroid gland[20] (Fig. 4d). Therefore, State 1 was termed as the follicular-like thyrocyte phenotype. (2) On the other hand, State 2 was basically a half-half mixture of tumor and LN-metastatic thyrocytes, displaying an intermediate state in malignant programs (Fig. 4c; Supplementary Fig. 5c). Consistent with the high proportion of LN metastases, State 2 thyrocytes had several features of EMT, including increased expression of extracellular matrix-related genes (*SDC4*, *ECM1*, *LGALS1*), upregulated EMT-related TFs (*HMGA2* and *EGR1*) and an enriched EMT signaling pathway (Fig. 4d, e; Supplementary Fig. 5d). Nonetheless, despite the downregulation of certain thyroid epithelial genes (*TPO*, *TFF3*, *DIO2*), it still maintained an overall expression of other epithelial markers (*TG*, *KRT18*, *EPCAM*) (Supplementary Fig. 5e, f). Furthermore, we did not detect other classical EMT TFs, such as TWIST1/2, ZEB1/2 and SNAIL1/2. Actually, this type of transcriptomic program did not support a full EMT, but reflected a biological process called partial EMT (p-EMT), which has been featured in multiple cancers[21–23]. Taken together, State 2 was termed as the p-EMT-like thyrocyte phenotype. (3) State 3, featured by the lowest TDS score and *RAS* score, and highest *BRAF* score, contained the vast majority of RAIR subcutaneous metastatic thyrocytes (Fig. 4c; Supplementary Fig. 5c, g, h). Meanwhile, State 3 thyrocytes had preferentially upregulated dedifferentiation-related TFs (*GATA2*, *MYC*, *SOX4*) and pathways (E2F_TARGETS, HYPOXIA, MYC_TARGETS_V1) (Fig. 4d, e), and simultaneously exhibited the lowest level of

thyroid epithelial markers (*TG*, *TPO*, *TFF3*, *DIO2*, *ID4*) (Supplementary Fig. 5e, f). Furthermore, we obtained the scRNA-seq data of anaplastic thyroid carcinoma (ATC) from the Gene Expression Omnibus (GEO, accession number: GSE148673) database and analyzed all the shared genes with our PTC profiles. We found that the Pearson's correlation between State 3 thyrocytes and ATC cells was as high as 0.72. Moreover, *TMSB4X*, a potential biomarker of PTC progression as described above, was also expressed at the highest level in State 3 (Supplementary Fig. 5i, j). Combining these observations, we termed State 3 as the dedifferentiation (dediff)-like thyrocyte phenotype.

Beyond the overall landscape, we then separately depicted the evolutionary paths of individual patients' thyrocytes. For single individuals in our scRNA-seq cohort, thyrocyte evolutionary dynamics well fit the lesion's clinicopathological characteristics. For example, (1) in Case 6, the right neck LN metastases (LN6r) had a low TDS score and were pathologically determined as TCV, a typical *BRAF*-like histologic subtype[4]. Accordingly, the LN6r cells were enriched in the dediff-like state with augmented *BRAF* signaling (Supplementary Fig. 6a). (2) Likewise, in Case 10, the advanced primary tumor (T10) invading thyroid cartilage contained more representative papillary architectures than the matched LN metastatic lesion (LN10r). In line with this pathological observation, T10 cells were closer to the evolutionary endpoint than LN10r cells on the trajectory (Supplementary Fig. 6b).

In addition, thyrocyte evolutionary dynamics of single individuals could also provide reasonable explanations for their different responses to RAI therapy. As mentioned above, thyrocytes from both two RAIR subcutaneous loci (SC4 and SC11) predominantly located in State 3 and presented very low TDS scores (Fig. 4f; Supplementary Fig. 6c), leading to a logical assumption regarding the strong relationship between RAIR disease and abundance of dediff-like cells. Notably, this hypothesis was backed up by the disease course of two FV-PTC cases (Case 5 and 7) in our cohort. (1) In Case 7, the right LN metastases (LN7r) which had prevalent and well-retained follicular architectures, mostly lay in the follicular-like and p-EMT-like states (Supplementary Fig. 6d). Despite the extensive nodal involvement in preoperative CT scans and postoperative pathological examinations (15/31 metastatic LNs), after two adjuvant RAI treatments, he had no signs of tumor recurrence with undetectable serum thyroglobulin (Tg) and negative whole-body RAI scans throughout the 12-month follow-up period. (2) Contrastingly, the situation of another FV-PTC patient (Case 5) seemed to be much tougher. This 15-year-old male, who had a large primary lesion with extensive neck and lung metastases at initial surgery, has so far experienced three postoperative RAI treatments with a cumulative dose of 400 mCi (Supplementary Fig. 7a, b). Despite a transient reduction in serum Tg after the first RAI administration, his Tg levels rapidly rebounded and exceeded the baseline values during the following two RAI treatments, accompanied by continuous radiographic progression of lung metastases (Supplementary Fig. 7a–c). On the developmental trajectory, we found this case harbored ample dediff-like cells (Supplementary Fig. 7d), which might account for his RAIR clinical course.

**Refined bulk molecular subtyping identifies a distinct *BRAF*-like subclass with worse prognosis and promising prospect of immunotherapy.** To explore the generality and prognostic significance of thyrocytes' scRNA-seq-derived signatures, we identified 480 pseudotime-associated genes (PAGs) through the correlations between gene expressions and pseudotime of each thyrocyte (Supplementary Data 4). As these genes purely

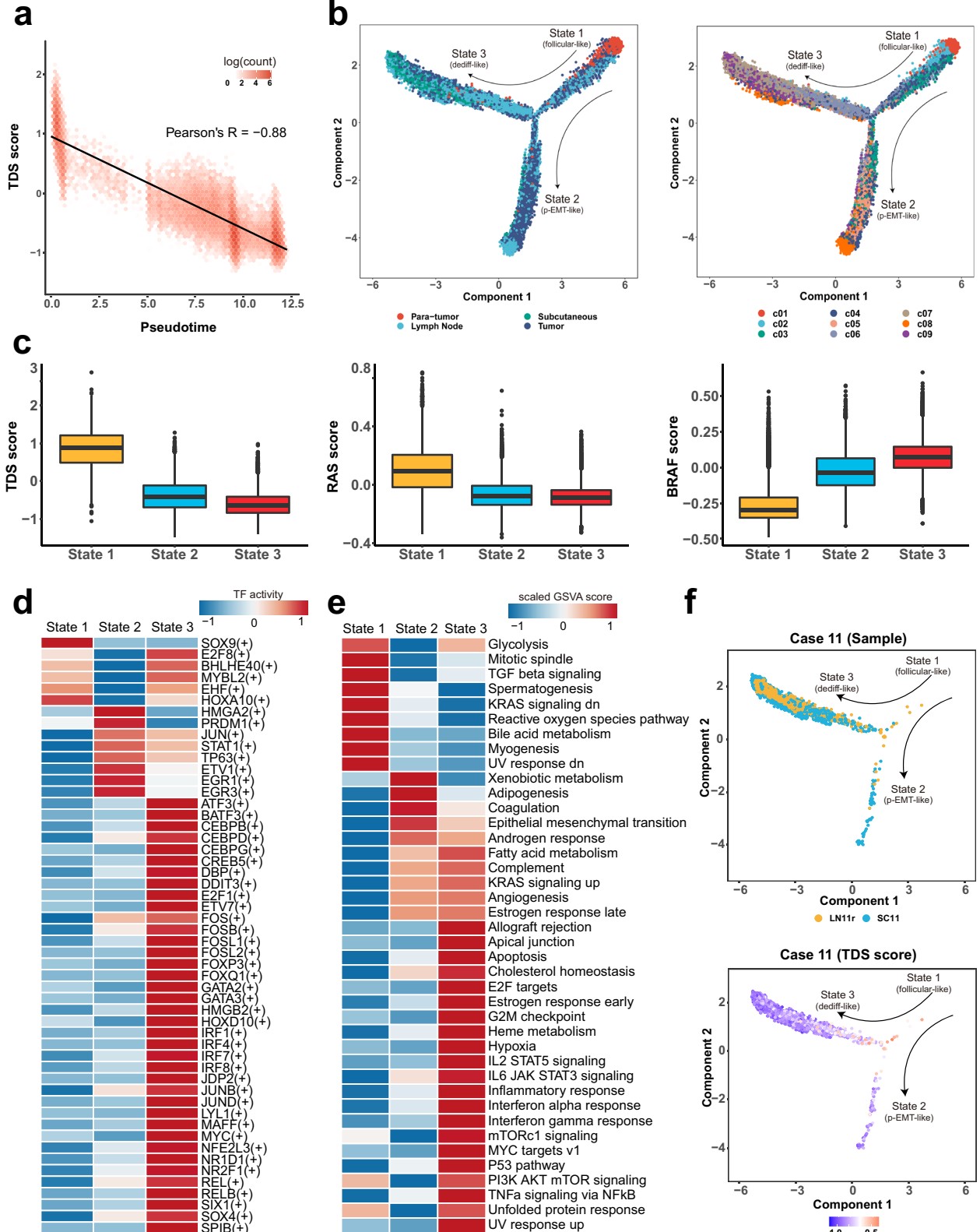

**Fig. 4 Trajectory analysis reveals three distinct transcriptional states of thyrocytes. a** Scatter plot showing a significantly negative correlation between TDS scores and pseudotime values inferred from the trajectory analysis for all thyrocytes. **b** Potential trajectory of all thyrocytes identified three distinct cell states (State 1–3), colored by tissue type (left) or cluster (right), respectively. The arrow shows potential evolutionary direction in the trajectory. **c** Boxplots showing significant differences in TDS scores, *RAS* scores and *BRAF* scores of the State 1 ($n = 7873$), State 2 ($n = 12,354$) and State 3 ($n = 16,038$) thyrocytes. The middle lines of the boxplots show the median (central line), the lower and upper hinges show the 25–75% interquartile range (IQR), and the whiskers extend from the hinge to the farthest data point within a maximum of 1.5x IQR. **d** Heatmap of TF (transcription factor) activities in the three thyrocyte states, scored by SCENIC. **e** Differences in the activities of hallmark pathways between different thyrocyte states, scored by GSVA. **f** Developmental trajectory of thyrocytes from Case 11, colored by tissue type (upper panel) or TDS score (bottom panel). Source data are provided in the Source data file.

represented the profiling of thyroid epithelial lineages rather than miscellaneous tumor components, we thus aimed to refine the conventional *BRAF*-/*RAS*-like molecular subtyping algorithm[4] based on PAGs using a cohort integrating two bulk RNA-seq profiles (TCGA and PRJEB11591, Supplementary Data 5; "Methods"). In general, our refined molecular classification changed the *BRAF*-like or *RAS*-like assignments of 6.9% (42/613) patients in the integrated bulk cohort, including *RAS*-driven tumors that were previously categorized into the *BRAF*-like subtype and vice versa (Fig. 5a; Supplementary Data 5).

Similar to the conventional bulk molecular classifications[4], the refined *BRAF*-like tumors also had strong heterogeneity in transcriptional outputs, while activation of thyroid hormone metabolism-related pathways was largely preserved in the refined *RAS*-like tumors (Fig. 5a; Supplementary Fig. 8a). These findings prompted us to dissect the transcriptional discrepancies of *BRAF*-like populations. Furthermore, unsupervised clustering classified the refined *BRAF*-like PTCs into two different subgroups (*BRAF*-like-A and *BRAF*-like-B), in which the *BRAF*-like-B subtype was not only associated with TCV pathology ($P = 1.7e-8$), lower TDS scores ($P = 2.2e-16$) and advanced staging ($P = 0.0008$) (Fig. 5b; Supplementary Fig. 8b, c), but also predicted a significantly compromised DFS ($P = 0.0059$) (Fig. 5c).

With regard to gene expression signatures, the *BRAF*-like-B subtype was characterized by higher activities of immune-related signalings, including but not limited to PD-1, interferon gamma (IFN-γ), major histocompatibility complex (MHC)-II antigen presenting pathways (Fig. 5d; Supplementary Fig. 8d). These findings suggested a promising therapeutic potential of immunotherapy for this subpopulation. In our scRNA-seq cohort, Case 10 represented a typical example of *BRAF*-like-B subtype who demonstrated an advanced primary tumor with prominent immune infiltration, positive expression of $BRAF^{V600E}$-mutant protein and programmed death-ligand 1 (PD-L1) protein (Fig. 5e), and further studies with more *BRAF*-like-B samples are required for further verification. Moreover, the predominant proportion of dediff-like cells in Case 10 (Supplementary Fig. 6b) raised a hypothesis that bulk molecular subtypes might be largely determined by the diversities in malignant thyrocyte phenotypes.

**Thyrocyte phenotype composition shapes the refined bulk classification.** To test the hypothesis above, we used the BisqueRNA approach to quantify the proportions of relevant cell types by deconvolution of the integrated bulk profiles[24]. Interestingly, the refined bulk subtypes (*RAS*-like, *BRAF*-like-A, *BRAF*-like-B) well corresponded to the abundance of three thyrocyte phenotypes (follicular-like, p-EMT-like, dediff-like). To be specific, (1) follicular-like thyrocyte was the preponderant phenotype in *RAS*-like tumors (Fig. 5f), which was concordant with the enrichment of FV-PTCs in this bulk subtype. (2) *BRAF*-like-A tumors were mainly composed of p-EMT-like thyrocytes (Fig. 5g), consistent with their relatively high rate of LN metastases (41.4%, Supplementary Data 5). Meanwhile, heterogeneity in tumor malignant degree might also be influenced by the proportion of dediff-like cells, as exemplified by Case 8 (17.5%, T4aN1bM0) and Case 9 (7.3%, T1bN1aM0) in our cohort (Supplementary Fig. 8e, f). (3) By contrast, *BRAF*-like-B tumors revealed an overwhelming dominance of dediff-like thyrocytes (Fig. 5h), potentially explaining the worse prognosis of this RAIR-prone subtype, such as Case 4 and Case 11 in our cohort (Fig. 4f; Supplementary Fig. 1a, 6c). Taken together, thyrocyte phenotype composition is an important factor that shapes our refined bulk classifications, and contributes to their clinicopathological and prognostic diversities.

**Cancer-associated fibroblast subtyping and their contributions to the PTC ecosystem.** Next, we turned our focus to stromal cells in PTC, among which cancer-associated fibroblasts (CAFs) act as key components in the TME with diverse functions[25]. In our scRNA-seq dataset, we found that all fibroblasts, regardless of tissue types, expressed the canonical CAF biomarkers, including *VIM*, *S100A4*, *ACTA2* (α-SMA), and *PDGRFA*, and were confidently defined as CAFs (Supplementary Fig. 9a, b). The CAFs were partitioned into four distinct clusters upon unsupervised t-SNE clustering, exhibiting either myofibroblastic or inflammatory phenotypes based on the mutually exclusive relationship of corresponding transcriptome signatures (Supplementary Fig. 9c–e). Inductively, cluster 0, 1 and 3 in Supplementary Fig. 9a were identified as myofibroblastic CAFs (myoCAF) due to the upregulation of canonical myofibroblastic markers including α-smooth muscle actin (αSMA, also called *ACTA2*) and contractile proteins (*TAGLN*, *MYLK*, *MYL9*) (Fig. 6a, b; Supplementary Fig. 9e). On the other hand, cluster 2 in Supplementary Fig. 9a represented the inflammatory subtype (iCAF) that preferentially expressed iCAF signatures such as *CFD*, *PLA2G2A*, *CCDC80* (Fig. 6a, b; Supplementary Fig. 9e). Of note, albeit to some shared upregulated immunomodulatory molecules, abundant expression of cytokines (such as *IL6*, *IL8*), which is a hallmark of iCAFs in pancreatic and breast cancers[26,27], was not observed in their counterparts of PTCs, suggesting heterogeneities in the iCAF-related regulatory mechanisms across different cancer types.

Subsequently, we deconvoluted the integrated bulk profiles (TCGA and PRJEB11591) to dissect the abundance of myoCAFs and iCAFs in our refined bulk subtypes. Consistent with previous reports[28,29], the *BRAF*-like subtype revealed a significantly higher fraction of CAFs than the *RAS*-like subtype, especially for *BRAF*-like-B tumors that were predicted to contain the peak CAF level regardless of myoCAF or iCAF phenotypes (Fig. 6c, d). To validate this finding, we reviewed the H&E-stained slides of Case 10, a typical *BRAF*-like-B PTC in our scRNA-seq cohort. Just in accordance, we observed significantly dense desmoplasia in several fields of the primary tumor (Supplementary Fig. 9f), which from a side confirmed the extensive presence of CAFs in the TME.

To further clarify the potential function of CAF phenotypes, we utilized CellPhoneDB[30] to infer cell-cell communications between iCAFs or myoCAFs with other cell types based on the relative abundance of ligand-receptor (L-R) pairs. Despite a deficiency in cytokine expressions, chemokine-mediated signalings took its place to maintain the immunomodulatory capabilities of iCAFs, implying their important roles of recruiting and crosstalking with diverse immune cells in the TME (Fig. 6e). For example, iCAFs were predicted to significantly interact with $CD8^+$ T, NK and tumor cells via the CCL5-ACKR4 complexes (Fig. 6e), suggesting that regulation of cellular immunity is an important function for iCAFs in PTC. Meanwhile, iCAFs might also participate in the process of innate immunity, supported by a moderate interplay between iCAFs and myeloid cells via CCL3L3-DPP4 interactions (Fig. 6e). Contrastingly, given the absence of significantly enriched L-R pairs, myoCAFs appeared to lack obvious intercellular crosstalk in the TME of PTC (Fig. 6f), indicating that myoCAFs tends to exert mechanical and chemical influence on tumor progression rather than through direct cell communications, as described in other solid tumors[31–33].

**Vascular-immune crosstalk in the PTC ecosystem raises the therapeutic potential of combined anti-angiogenic and immunotherapy.** To gain further insight into tumor stroma, we next explored the transcriptional programs of another important stromal element, the endothelial cells (ECs). By integrating the

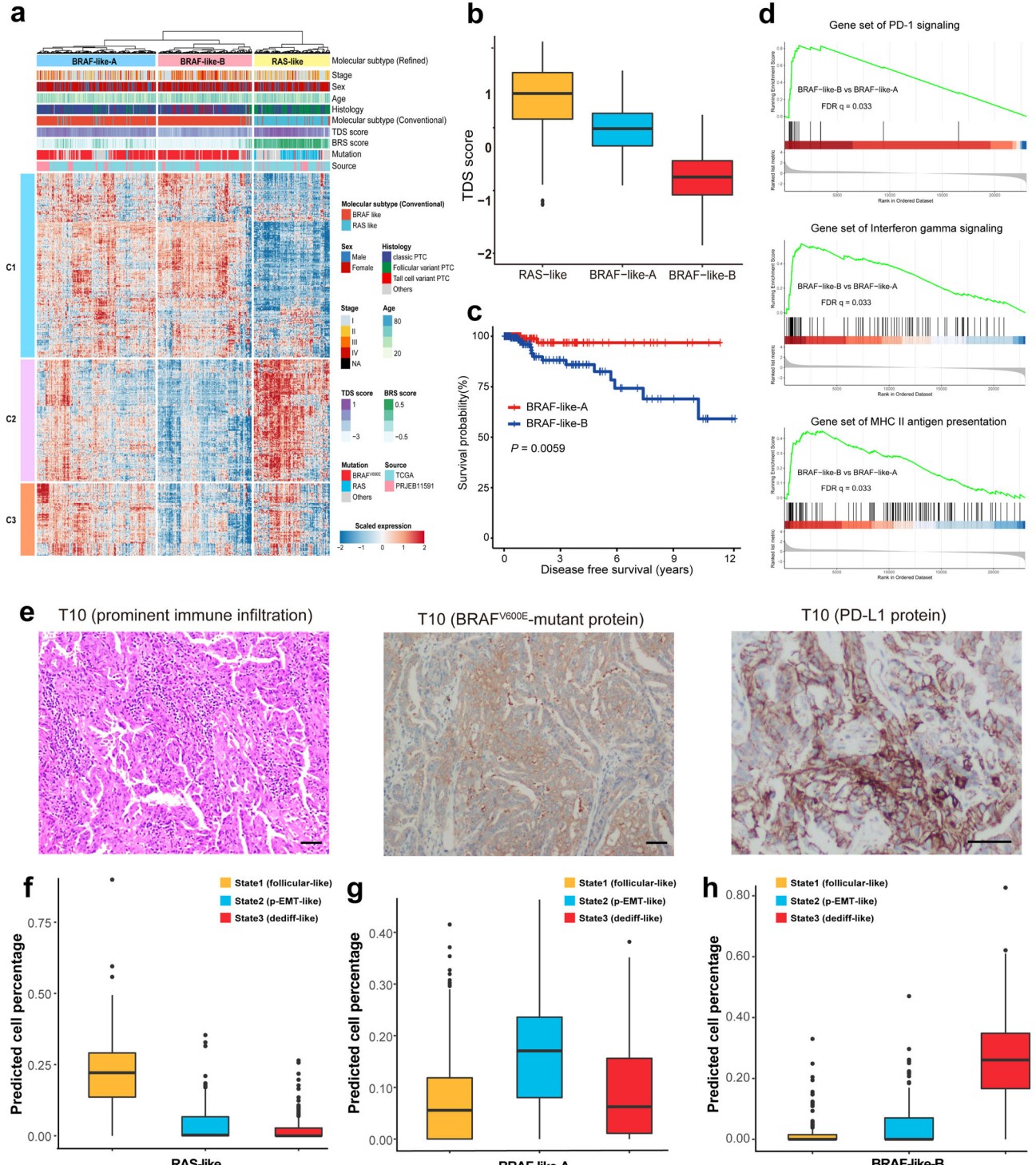

**Fig. 5 Modified molecular subtypes and deconvolution of bulk RNA-seq profiles. a** Heatmap of the top pseudotime-associated genes (*n* = 480) in 613 PTCs from the integrated bulk cohort (TCGA and PRJEB11591). Hierarchical clustering identifies three molecular subtypes of PTCs. **b** Differences in TDS scores of the three PTC molecular subtypes, *RAS*-like (*n* = 161), *BRAF*-like-A (*n* = 253) and *BRAF*-like-B (*n* = 199) defined in our study. **c** Kaplan–Meier plot for disease-free survival of patients with *BRAF*-like-A (*n* = 191) and *BRAF*-like-B (*n* = 171) PTCs in the TCGA cohort. Log-rank test (two-sided). **d** GSEA plots showing significantly enriched pathways in the *BRAF*-like-B subtype compared with the *BRAF*-like-A subtype. **e** H&E-stained or IHC images showing prominent immune infiltration, positive BRAF V600E-mutant protein and PD-L1 protein expressions in the primary tumor (T10) of Case 10. Three independent experiments were performed and generated similar results. Scale bar = 50 μm. Deconvolution analysis showing thyrocyte phenotype composition in the (**f**) *RAS*-like subtype (*n* = 161), (**g**) *BRAF*-like-A subtype (*n* = 253) and (**h**) *BRAF*-like-B subtype (*n* = 199). In (**b**), (**f**), (**g**) and (**h**), the middle lines of the boxplots show the median (central line), the lower and upper hinges show the 25–75% interquartile range (IQR), and the whiskers extend from the hinge to the farthest data point within a maximum of 1.5 x IQR. Source data are provided in the Source data file.

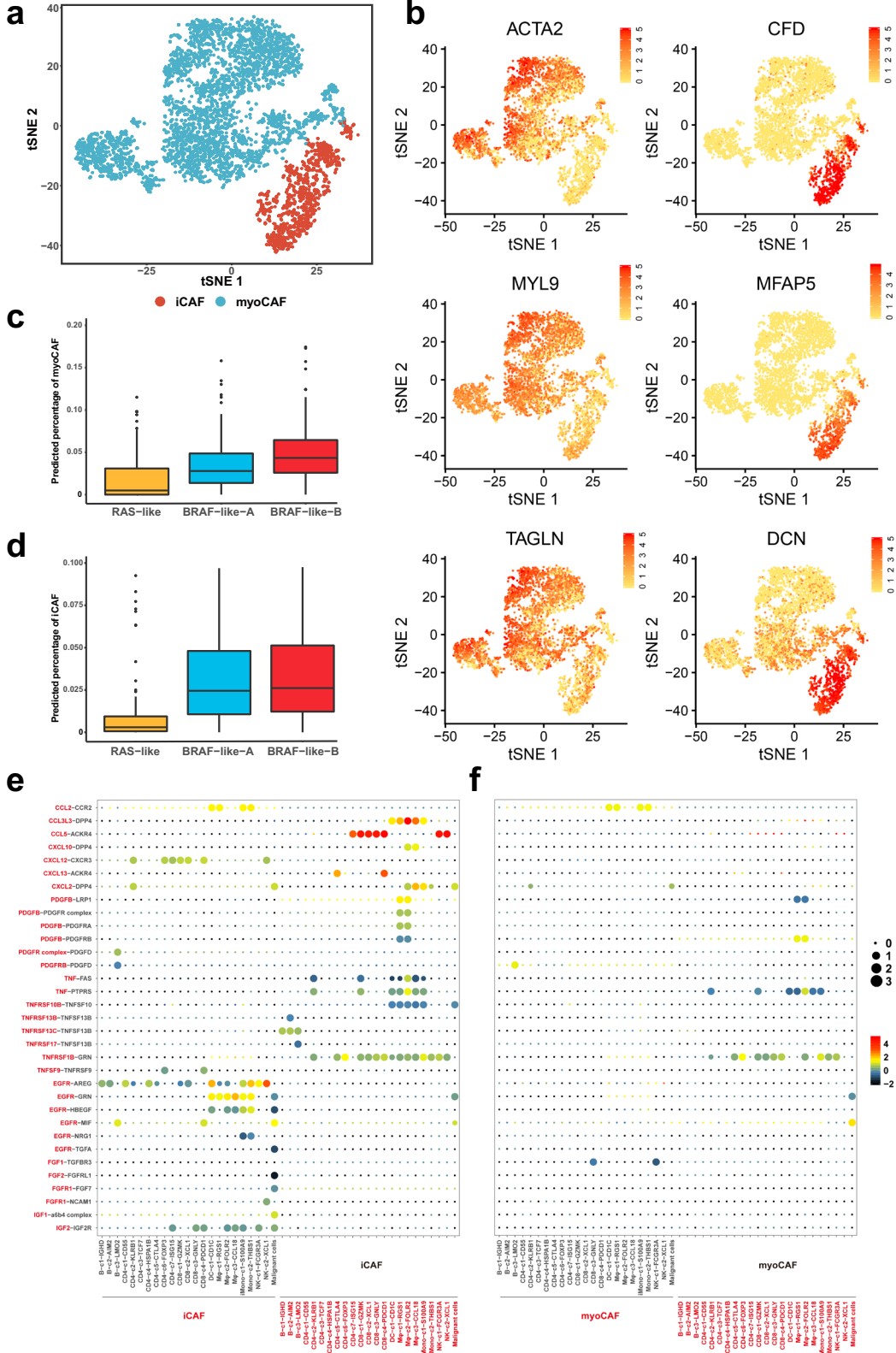

**Fig. 6 Subtyping of cancer-associated fibroblasts and their contributions to the PTC ecosystem. a** t-SNE projection of the distinctions between myoCAFs and iCAFs. **b** t-SNE plots of key marker genes for myoCAFs and iCAFs in PTC. Deconvolution analysis demonstrating the predicted proportion of (**c**) myoCAFs and (**d**) iCAFs in the three bulk PTC classifications (*RAS*-like (*n* = 161), *BRAF*-like-A (*n* = 253), *BRAF*-like-B (*n* = 199)). The middle lines of the boxplots show the median (central line), the lower and upper hinges show the 25–75% interquartile range (IQR), and the whiskers extend from the hinge to the farthest data point within a maximum of 1.5x IQR. Bubble plots showing ligand-receptor pairs of cytokines, chemokines and growth factors between (**e**) iCAFs or (**f**) myoCAFs with other cell types, inferred by CellPhoneDB. Source data are provided in the Source data file.

function-related gene signatures in a study of lung cancer ECs[34], we classified these ECs into arterial, venous, lymphatic, immature and tip phenotypes (Fig. 7a). Just as the name implies, arterial ECs expressed high level of genes associated with arterial development and remodeling (*FBLN5*, *GJA5*, *JAG1*) and smooth muscle contraction (*PPP1R14A*) (Fig. 7b), while the venous subtype revealed upregulation of *VWF* (von Willebrand factor) and genes associated with leukocyte recruitment (*ACKR1*) or adhesion (*SELE*) (Fig. 7c). Lymphatic ECs were enriched for a canonical marker *LYVE1* and a chemokine ligand *CCL21* (Fig. 7d), while immature ECs were characterized by a higher expression level of Notch signaling and target genes (*JAG1*, *HES1*, *ID1*, *ID2*, *ID3*), and genes involving in barrier integrity (*ENG*, *PLVAP*, *HSPG2*, *APLNR*), which may resemble the stalk-like cells (Fig. 7e).

Remarkably, tip ECs, the key phenotype in sprouting angiogenesis, expressed a signature of genes involved in cell migration (*NRP1*, *ENPP2*), adhesion (*THY1*) and vessel formation (*FLT1*, also called *VEGFR1*; *KDR*, also called *VEGFR2*; *NRP1*, also called *VEGF165R*) (Fig. 7f), facilitating their navigating role in the process of vessel sprouting. In agreement with these signatures, SCENIC analysis identified several upregulated TF regulons in tip cells that have been reported to be related with endothelial migration and sprouting, such as *ZEB1*, *HOXB5* and *STAT* family (*STAT1*, *STAT2*)[35–37] (Fig. 7g), further corroborating the pro-angiogenic properties of tip phenotype in PTCs. Moreover, with regard to cell origins, almost all the tip, arterial and immature ECs were found to be located in primary or metastatic tumor samples, while only lymphatic ECs were enriched in normal thyroid tissues (Fig. 7h). Collectively, these observations suggested that vasculogenesis serves as an important hallmark of PTCs, and the tip EC phenotype is likely to be a promising target of anti-angiogenic therapy (AAT), especially anti-vascular endothelial growth factor receptor (VEGFR) antibodies in this disease.

To dissect the complex activities of tip cells in PTCs, we subsequently used CellPhoneDB to investigate the molecular communication networks of this phenotype. In this analysis, we observed widespread interactions between ECs and immune cells. For instance, we found that the lymphatic ECs interacted with immune cells through the atypical chemokine receptor 2 (*ACKR2*), which has been reported to regulate chemokine availability (Supplementary Fig. 10)[38]. Meanwhile, we found that venous, immature and arterial ECs interacted with immune cells through the Intercellular Adhesion Molecule 1 (*ICAM1*) on its surface, while the *ICAM1* expression was significantly reduced in tip cells and lymphatic ECs (Supplementary Fig. 11). In particular, we identified that the crosstalks between tip ECs and immune cells were predominantly achieved through the key angiogenic VEGF-VEGFR signalings (Fig. 7i; Supplementary Fig. 12). Taken together, these results supported the presence of extensive vascular-immune crosstalk in the multicellular tumor ecosystems[39]. Therefore, these observations raise the therapeutic potential for AAT alone or in combination with ICB in thyroid cancers.

## Discussion
Cancer is not a quiescent disease, but it has only recently become possible to study the evolving patterns of cancer cells and surrounding stromal cells with maturity and extensive application of single-cell sequencing technologies[40,41]. However, up to now, there is still a lack of relevant research in thyroid cancers. In this study, we carried out scRNA-seq analysis covering paratumor, localized/advanced tumors, initially-treated/recurrent LNs and RAIR distant metastases from PTC patients with diverse clinical courses. Overall, our single-cell data have considerably improved

our understanding of PTC heterogeneity and added dimensions to prognostic stratification and tailored therapeutics (Fig. 8).

In previous scRNA-seq studies of many other cancers[22,42–45], identification of malignant and normal compartments largely relies on the inferred large-scale copy-number variation (CNV) analysis of single cells, provided that cancer cells commonly harbor chromosomal changes in these malignancies. Nonetheless, the low frequency of somatic CNVs (27.2%) of PTCs[4] precludes the application of CNV-based classification in thyroid cancers. To address this challenge, we established a set of integrated approaches combining tissue origins, transcriptome correlations, TDS-related transcriptome signatures and a machine learning classifier to distinguish malignant thyrocytes from their non-malignant counterparts in PTCs with a high discriminative power. In this way, our study provides a referable means to ascertain malignant thyrocytes in future scRNA-seq studies.

One of the key findings in our study is the identification of premalignant thyrocytes. A growing body of evidence has focused on the aberrant transcriptional profiles of normal tissues adjacent to the tumor (NAT) in breast, prostate, colon and liver cancers[46–49]. In particular, a recent pan-cancer analysis also confirmed the intermediate state of NAT between non-tumor-bearing healthy tissues and tumor samples across various cancer types[50]. In this study, we discovered two distinct types of histologically normal thyroid epithelial cells. The major type possibly represents the truly normal thyrocytes, while the other population represents the premalignant thyrocytes with normal morphology but intermediate transcriptomes between normal and cancer cells. Most importantly, we confirmed the cancer-primed properties of premalignant cells that independently give birth to cancer cells with even different histology from the first primary lesion (Fig. 3d). The existence of premalignant thyrocytes can be explained by the "field cancerization" theory that genetic alterations of tumor accumulate in a stepwise manner, accompanied by the occurrence of cancer-primed cells that may show no morphological change before cancer formation[51]. Although the universality of the premalignant thyrocytes needs to be further validated in future studies, these observations and hypotheses, to some extent at least, provide a mechanistic insight into the multicenter onset of tumorigenesis in a fraction of PTCs.

As mentioned above, our study was highlighted by the collection of a broad spectrum of tissue origins, facilitating us to delineate comprehensive evolutional paths of thyroid epithelial cells along with cancer progression. Our study defines three phenotypes of thyrocytes (follicular-like, p-EMT-like and dediff-like) at single-cell resolution, and further outlines two divergent routes from follicular-like to either p-EMT-like or dediff-like thyrocytes, where the thyrocyte positions on the trajectory fit well with tumor characteristics and RAI responses for each individual case in our cohort. Meanwhile, Knauf et al. found that PTC could further develop into poorly-differentiated thyroid cancer (PDTC) through MAPK-dependent EMT process. This important finding is partly consistent with the development of p-EMT/dediff-like thyrocytes in our study, as KRAS signaling of State 2/3 thyrocytes was activated, which in turn activated the MAPK pathway[52]. In line with other cancers[53], almost all tumors in our cohort are comprised of different thyrocyte phenotypes rather than a single clone with uniform traits, suggesting that dynamic evolution of cancer cells is ubiquitous and is largely responsible for intra-tumor heterogeneities in PTCs.

At the bulk level, the TCGA study represents a landmark of this field, classifying PTC into either a *BRAF*-like or a *RAS*-like molecular subtype according to the bulk profiles[4]. On this basis, we used the unique evolution-related genes of thyroid epithelial lineage to slightly modify the TCGA subtyping algorithm. More importantly, we uncovered a strong connection between bulk

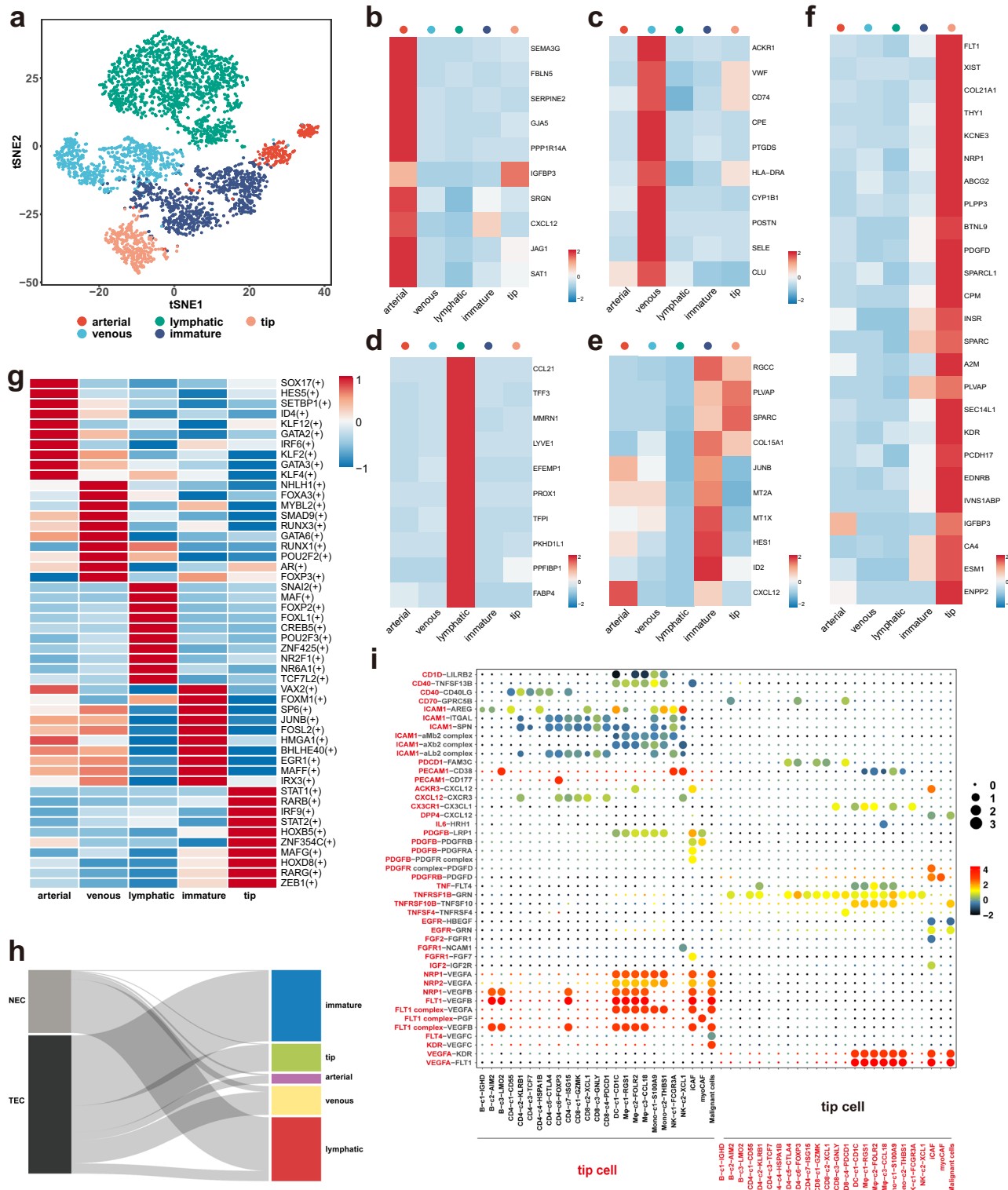

**Fig. 7 Characterization of endothelial cells in PTCs. a** t-SNE projection showing five different subtypes of ECs. Expression levels of marker genes for arterial (**b**), venous (**c**), lymphatic (**d**), immature (**e**), and tip ECs (**f**). **g** Heatmap showing the TF activities in the five EC subtypes, scored by SCENIC and AUCell. **h** Sankey diagram showing assignment of normal EC (NEC) and tumor EC (TEC) to arterial, venous, lymphatic, immature and tip subtypes. **i** Bubble plots showing L-R pairs of cytokines, chemokines and growth factors between tip ECs and other cell types, inferred by CellPhoneDB. Source data are provided in the Source data file.

molecular subtypes and abundance of different thyrocyte phenotypes, implying that the composition of different thyrocytes also shapes inter-tumor heterogeneities in PTCs. Among these bulk classifications, the *BRAF*-like-B subpopulation merits future

attention. Due to its almost pure composition of dediff-like cells, this subtype is prone to an aggressive and RAIR clinical course, thus results in a worse prognosis. Nevertheless, the enriched inflammatory (IFN-γ, MHC-II, etc.) and immunosuppressive

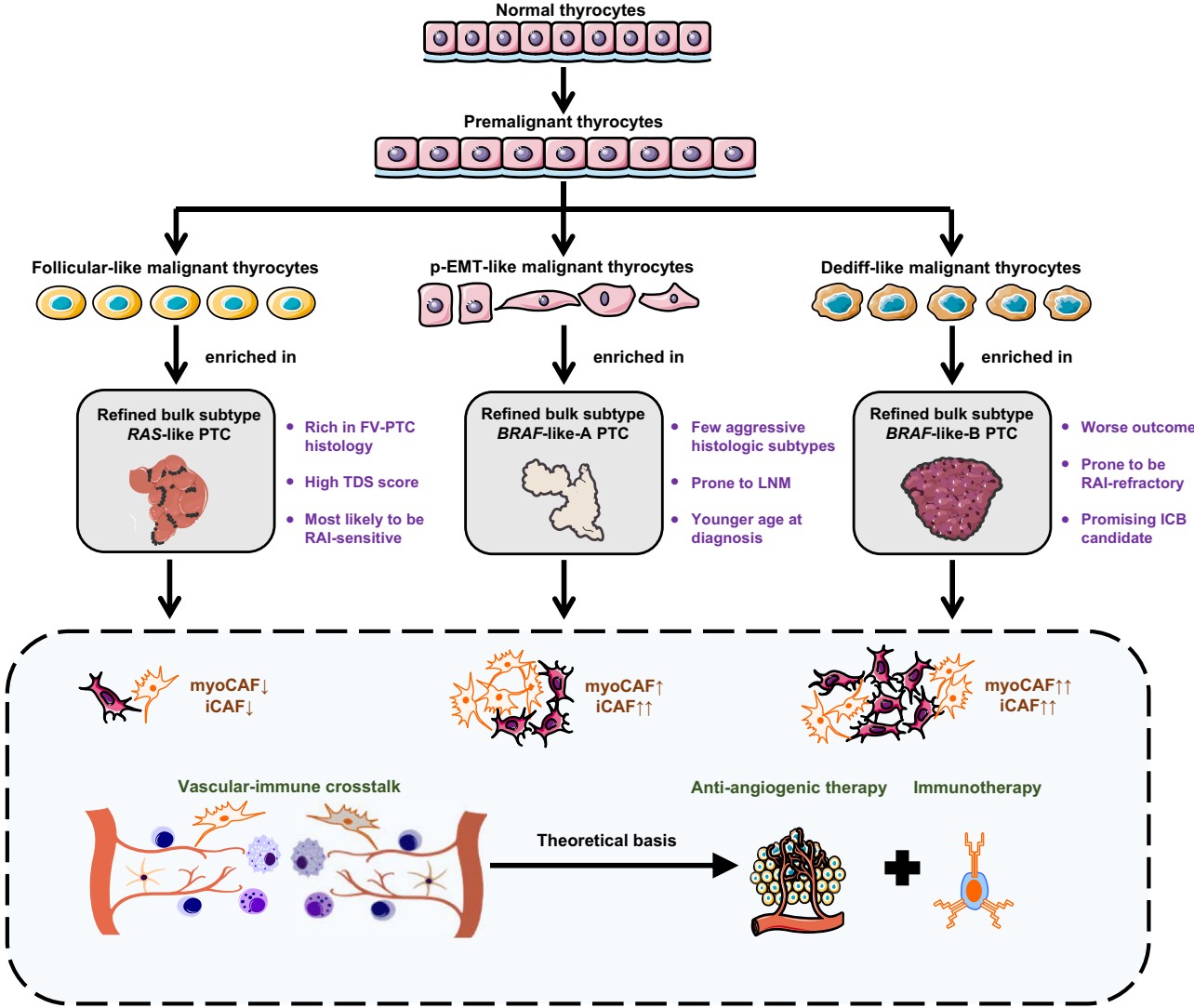

**Fig. 8 Schematic Digram of the present study.** Schematic Diagram Dissecting the PTC Ecosystem with Potential Prognostic and Therapeutic Implications.

(PD-1) signalings may bring hope to advanced *BRAF*-like-B patients who are potentially to be promising candidates for ICB. A few prior studies have reported partial or even complete regression of PTCs during the administration of PD-1 inhibitors[3,54,55], indicating that at least a subset of patients would really benefit from ICB.

The heterogeneity of stromal cells has been scarcely characterized in PTCs. Despite an abundance of literature supporting a tumor-promoting role of CAFs, they are now considered to be a heterogeneous entity that plays a very complex role in cancer[56]. In the present study, composition of both CAF phenotypes increased with the malignant degree of bulk classifications, suggesting that both phenotypes, or at least a substantial proportion, might be pro-tumorous. It is generally acknowledged that CAFs confer resistance to anticancer drug therapies including chemotherapy and tyrosine kinase inhibitors[33]. However, whether they mediate resistance to RAI (especially myoCAFs) or immunotherapy (especially iCAFs) in PTC remains unclear, and needs to be clarified by more focused studies.

Among the relevant findings in ECs, the tip phenotype can be a worthwhile focus. Due to its essential role in angiogenesis and preferential upregulation of VEGFRs, the anti-VEGFR therapy, to some extent, refers to anti-tip-cell therapy[57]. For example, the efficacy of sorafenib, the first multi-kinase inhibitor (including

VEGFRs) approved to treat RAIR differentiated thyroid cancer, has been shown to be closely correlated with a tip EC marker CXCR4 in hepatocellular carcinoma[58]. In addition, tip ECs were speculated to have widespread interactions with immunologic components (including immune cells and iCAFs) in this study. In line with our finding in PTCs, inhibition of VEGFR is known to remodel the tumor immune microenvironment[59], constituting the theoretical basis of combined anti-VEGFR therapy and ICB in the treatment of advanced thyroid cancers[60,61].

It is estimated that nearly 50% of the substantial increase in thyroid cancer incidence is attributed to papillary thyroid microcarcinoma (PTMC)[62,63], which means the maximal PTC foci measuring 1 cm or less. However, regardless of single-cell or bulk transcriptome profiles (such as TCGA), PTMC is inevitably underrepresented due to the limited tissue volume available for sequencing, especially for those tumors less than 5 mm, which would temper our conclusions to some extent. In the future, with advances in microsampling sequencing technologies, further study is needed to better parse the tumor ecosystems in this entity with a rapid-growing morbidity.

We acknowledge that our study has several limitations. First, the sample size and cell number in our study is limited, and further expansion of the study cohort is required to better elucidate the tumor progression and diversity in PTC. In particular,

the premalignant cells found in our study are primarily determined by one patient, and this state of thyrocytes should be verified in further studies. Second, current scRNA-seq must be performed within a short period of time after sample acquisition because fresh samples with viable cells are needed in this technology. It is still a challenge to completely correct the batch effects while fully retaining the true biological properties caused by inter-tumor heterogeneities. Though we performed the MNN algorithm for batch effect correction, batch effects may still exist in our study. Third, our study only examined the mutational status of bulk samples, while the genetic annotation of different thyrocyte clusters is lacking. Future studies with integration of multimodal single-cell sequencing data may delineate the DNA mutation spectrum, epigenetic modifications and transcriptome at the single-cell resolution simultaneously for different cell clusters. At last, our study performed a series of bioinformatic tools to dissect the TME of PTC, while the functional validation of these findings is lacking, such as the iodine uptake and retention in different kinds of thyrocytes and the roles of the identified TDS-associated genes in PTC tumorigenesis. Further in vivo and in vitro experiments are awaited to validate these findings in the future.

In summary, our study illuminates a comprehensive landscape of PTC ecosystem that suggests potential prognostic and therapeutic implications. Our work not only adds dimensions to the biological underpinnings of PTC heterogeneity, but also provides a benchmark dataset for this malignancy. We anticipate our data will serve as a valuable resource facilitating future studies to further develop biomarkers or treatment targets for PTC patients.

## Methods

**Ethical statement**. This study was reviewed and approved by the Institutional Review Board of Fudan University Shanghai Cancer Center. Written informed consent was obtained from each patient prior to sample collection.

**Human specimens**. Eleven patients who underwent surgery at Fudan University Shanghai Cancer Center (FUSCC) were included in our study. In total, 23 fresh surgical specimens (7 primary tumors, 6 para-tumors, 8 metastatic LNs and 2 subcutaneous metastatic loci) were sequenced and incorporated in further analyses. Hematoxylin and eosin (HE)-stained sections of each sample were carefully reviewed by two experienced pathologists to confirm the pathology. Clinical information including demographics, tumor clinicopathologic characteristics, treatment history and results of preoperative thyroid function test were summarized in Supplementary Data 1. Single-cell data information of the 23 samples is shown in Supplementary Table 2.

**Sample preparation**. Fresh samples were resected and washed twice with 1x PBS (Gibco). Each sample was cut into ~1 mm³ pieces and enzymatically digested with 10 mL digestion medium containing 1 mg/mL collagenase and 1 U/mL Dispase II (Gibco). These samples were subsequently incubated at 37 °C for 50 min and triturated with pipette per 15 min. The suspended cells were washed with 20% fetal bovine serum (FBS) in Dulbecco's Modified Eagle Medium (DMEM) and filtered through a 40-μm Cell-Strainer nylon mesh (BD) and centrifuged at 700 × g for 10 min. After removing the supernatant, the cell pellet was washed twice with MACS buffer (PBS containing 1% FBS, 0.5% EDTA, and 0.05% gentamycin) and then re-suspended in the sorting buffer (PBS supplemented with 1% FBS).

Subsequently, the suspended cells were stained with DRAQ5 and DAPI (Sigma) to harvest nucleated living cells. After antibody incubation, the cells were washed twice with cold PBS and reconstituted in DMEM with 20% FBS. Cell sorting was performed with a MoFlo Astrios EQ Cell Sorter (Beckman Coulter). Unstained cells were routinely used to define FACS gating parameters and sorted into DMEM media supplemented with 20% FBS.

**Whole-exome sequencing and Sanger sequencing**. Genomic DNA samples were extracted by GeneRead DNA FFPE Kit (QIAGEN,. Hilden, Germany) from formalin-fixed, paraffin-embedded (FFPE) tumor tissues from all 11 PTC patients included in our study. Then the DNA libraries were prepared and captured using Agilent SureSelect Human All Exon v6 kit following the manufacturer's protocol (Agilent Technologies, USA). The libraries were then sequenced on an Illumina NovaSeq 6000 sequencing platform (Illumina, Inc., San Diego, USA) and 150 bp paired-end reads were generated.

In brief, the raw reads were pre-processed with fastp (version: 0.19.5)[64]. After that, clean reads were aligned to the reference human genome (GRCh37) utilizing the BWA (version 0.7.12), which were then sorted and indexed by SAMtools (version 1.4)[65]. The final BAM files were used as input files for variant calling. The GATK (version 4.1.0.0) was used for recalibration of the base quality score and for single nucleotide polymorphism (SNP) and insertion/deletion (INDEL) realignment[66].

Regular PCR procedures were used to amplify *BRAF* and *RAS*, and nest-PCR for *TERT* promoter region. The following conditions were used for amplification: 1 cycle at 94 °C for 3 min, 35 cycles at 94 °C for 30 s, 59 °C (for *TERT* promoter) or 55 °C (for *BRAF* and *RAS*) for 30 s and 72 °C for 30 s, 1 cycle at 72 °C for 10 min[67]. The Sanger sequencing was performed using an ABI 3730XL analyzer. The sequences of the primers used were presented in Supplementary Table 12. Samples with mutation rates above 15% were deemed to have undergone mutation.

**Single-cell RNA-seq and reads processing**. The cell suspension of each sample was subjected to the Gel Bead Kit V3 (10x Genomics, Pleasanton, CA) for library preparation according to the standard protocols. The single cell libraries were sequenced on Illumina NovaSeq 6000 Systems using paired-end sequencing (150 nt). The gene-barcode matrices were generated by Cell Ranger toolkit (v3.1), which aligned the droplet-based sequencing data against GRCh38 human reference genome and counted the unique molecular identifiers (UMIs) for each cell.

**Quality control and batch effect correction of scRNA-seq data**. The "Seurat" R package (v3.1.4) was primarily applied for quality control procedures and downstream bioinformatic analyses[68]. We first filtered out cells with low quality that fit any of the following criteria: the proportion of mitochondrial genes counts (>10%), UMIs < 500 or UMIs > 5000. The DoubletFinder (v2.0) package was utilized to remove the potential doublets with the default settings[69]. After these quality control procedures, we conducted a series of preprocessing procedures for downstream analysis. In detail, we employed a global-scaling normalization method "Log-Normalize" that normalized the feature expression for each cell by the total expression and multiplied this by a scale factor (10,000 by default), and log-transformed the result using the NormalizeData() function in Seurat. After that, the normalized expression profiles of all samples were merged together using the merge() function in R v3.6.3. To correct batch effects, we conducted the matching mutual nearest neighbors (MNN) correction[70], and used its faster implementation in Python (mnnpy, v0.1.9.3). The top 5000 highly variable genes (HVGs) of the merged dataset identified by the FindVariableFeatures() function were utilized as input for batch effect correction. Finally, we obtained the scaled and batch effect-corrected expression profiles of all samples for downstream analyses.

**Unsupervised clustering and dimensional reduction**. The top principal components (PCs) were computed based on the gene expression profiles of the top 5000 HVGs after batch effect correction. The PCElbowPlot() function in Seurat was utilized to select the optimal number of PCs for further analysis as recommended by Seurat (v3.1.4). The FindNeighbors() and FindClusters() functions in Seurat were both applied for cell clustering. To find the optimal cluster resolution, a visualization-based method "clustering tree" was applied when required[71]. The RunTSNE() and RunUMAP() function were both performed for visualization when appropriate. The cell identity of each cluster was defined based on the expression of known marker genes. In the first round of "low-resolution" clustering, we identified myeloid cells (*LYZ, FCER1G, LYZ, TYROBP*), T and NK cells (*CD3D, CD3E, IL7R, IL32, TRAC*), B cells (*CD79A, CD79B, MS4A1, IGKC, CD74*), thyroid epithelial cells (also called thyrocyte, *TG, CLU, FN1, MGST1, S100A13*), fibroblasts (*RGS5, IGFBP7, TAGLN, COL1A2, ACTA2*) and endothelial cells (*TIMP3, RAMP2, CLDN5, TFPI, MGP*). Due to the transcriptional similarities between NK and T cells, we conducted the second round of clustering to distinguish T and NK cells. After that, we conducted the third round of "high-resolution" clustering to identify the finer subclusters within each major cell types. Procedures of the second and third round of clustering were identical to the first one, all starting from the computation of PCs and then clustering cells with the optimal resolution obtained from the "clustering tree" method to seek the identity for each subcluster.

**Identification of signature genes for cell clusters**. The differential expressed genes (DEGs) in each subcluster were identified through the FindAllMarkers() function in Seurat. Significance levels of these signature genes were determined using the Wilcoxon rank-sum test along with Bonferroni correction. The signature genes of each cluster were determined based on the following criteria: (1) expressed in more than 20% of the cells within either or both two groups; (2) |log₂FC| > 0.5; (3) Wilcoxon rank-sum test adjusted *P* value < 0.01.

**Immunohistochemical staining**. Formalin-fixed, paraffin-embedded (FFPE) tumor and paratumor tissues of two additional cases (classical PTC and FV-PTC) were separately sliced into 4-μm sections and mounted on glass slides. The slides were baked at 65 °C overnight. After deparaffinization and hydration, these slides were boiled in citrate buffer at 100 °C for 15 min. Subsequently, a 3% H₂O₂ solution was used to block endogenous peroxidase activities for 20 min. To prevent

nonspecific antibody binding, the slides were then incubated with 5% normal goat serum for 1 h at room temperature. Then these slides were incubated at 4 °C overnight with anti-TMSB4X primary antibody (Proteintech, 19850-1-AP), 1:100, which was validated for IHC by the manufacturer on mouse skeletal muscle tissue and colon tissue. Following washes with TBST for 3 times, the slides were then incubated with HRP-conjugated goat anti-rabbit/mouse secondary antibody (GeneTech, GK500705) for 1 h at room temperature. Sections were stained by DAB and then counterstained with hematoxylin according to the manufacturer's instructions.

For the primary tumor of Case 10 (T10), IHC staining of $BRAF^{V600E}$-mutated protein (Ventana BRAF V600E [VE1] antibody, 0786227000) and PD-L1 protein (Dako 22C3 pharmDx assay, SK006) was conducted by the Department of Pathology at FUSCC on a Dako Autostainer Link 48 system (Agilent). Both of these two assays are in vitro diagnostic products approved by the US Food and Drug Administration (FDA). The subsequent procedures were identical to that of TMSB4X staining.

**Inference of cell state by trajectory analysis.** The trajectory analysis was performed using the Monocle2 package (v2.14.0) to reveal the cell-state transitions[72]. The ordering genes in the trajectory analysis were determined according to each gene's expression (mean expression > 1) and variance level (dispersion_empirical > 2 × dispersion_fit) as recommended by Monocle2. The DDRTree() function in Monocle2 was applied to reduce the dimensions with default settings. As the para-tumor thyrocytes have been separated and clearly defined in our study, we set these cells as the root state and performed the "order" function in Monocle2. The DEGs changed along with the pseudotime were identified using the differentialGeneTest() function in Monocle2.

**Analysis of bulk RNA-seq data.** The bulk RNA-seq profiles were integrated from both the TCGA dataset and the PRJEB11591 dataset (available in EBI European Nucleotide Archive database with accession number PRJEB11591)[19], in which the transcript reads were processed and quantified by kallisto (v0.46.1) against the annotated transcripts (Gencode v24)[73]. The gene-level expression of each sample was calculated by aggregating transcript expression (calculated by Transcripts Per Million, TPM) belonging to the same gene with the "tximport" package (v1.14.0). The DEGs between tumors and para-tumors were calculated by DESeq2[74]. To refine the bulk molecular subtyping of PTCs, we selected the top candidate genes ($n = 480$) which changed along with the pseudotime of the thyrocytes in the trajectory analysis. The heatmap was then plotted by the "pheatmap" package (v1.0.12) in R for visualization. The Gene Ontology (GO) and Kyoto Encyclopedia of Genes and Genomes (KEGG) enrichment analyses were completed using the "clusterProfiler" package (v3.14.2) in R[75]. Survival analysis was performed using the "survival" package (v2.44) in R and the GEPIA2 web-based tool (http://gepia2.cancer-pku.cn/)[76].

The previous TCGA study has developed a continuous thyroid differentiation score (TDS) using the expression profiles of 16 thyroid function-related genes to quantify relationships between thyroid differentiation and diverse genetic or epigenetic events[4]. Among the 16 genes, three genes encoding microRNAs were unavailable in the PRJEB11591 dataset, and thus were removed in our further analysis. Collectively, we first log-normalized and scaled the expression levels of the 13 mRNA genes (TG, TPO, SLC26A4, DIO2, TSHR, PAX8, DUOX1, DUOX2, NKX2-1, GLIS3, FOXE1, TFF3, FHL1) by centering them at the median expression value. The TDS score was calculated as the sum of log2 (fold change) across these 13 genes.

In addition, the TCGA study has investigated transcriptional diversities and separated PTCs into $BRAF$-like and $RAS$-like[4]. In accordance with this study, we used the expression profiles of 71 DEGs between $BRAF^{V600E}$ and $RAS$ mutated samples, and quantified whether a given tumor resembled either the $BRAF$-like or $RAS$-like group. In detail, we first computed the centroids of the 71-gene signature in $BRAF^{V600E}$-mutated samples as $c(B)$, and the $RAS$-mutated samples as $c(R)$. The 71-gene signature of each tumor (t) was represented as $v(t)$. The $BRAF^{V600E}$-$RAS$ score (BRS) of each tumor (t) was then defined as the difference between the normalized Euclidean distance of $v(t)$ from $c(B)$ and $c(R)$:

$$BRS(t) = |v(t) - c(B)|^2 - |v(t) - c(R)|^2 \qquad (1)$$

Tumors with negative BRS were considered as $BRAF$-like, while those with positive BRS were defined as $RAS$-like.

**Construction of TDS, $BRAF$ and $RAS$ scores of thyrocytes in the scRNA-seq dataset.** Seurat's AddModuleScore function was applied to quantify differentiation status of each thyrocyte in our scRNA-seq dataset. We calculated the mean abundance levels of the abovementioned 13 thyroid function-related mRNA genes against the aggregated abundance of random control gene sets as the TDS score of each thyrocyte. Similarly, the $BRAF$ and $RAS$ score of each thyrocyte were quantified with AddModuleScore() function based on the upregulated signature genes in the $BRAF^{V600E}$-mutated and $RAS$-mutated samples from the bulk TCGA profiles (Supplementary Data 6).

**Prediction of malignant thyrocyte identity based on bulk profiles.** To distinguish malignant compartments from the acquired thyrocytes, we proposed the

K-nearest neighbors (KNN) classification method in conjunction with Spearman's correlation coefficient as distance measurement. In detail, we first labeled the TCGA tumor/para-tumor samples as malignant/non-malignant. Next, we computed Spearman's correlation between each thyrocyte and each sample in the TCGA dataset using highly variable genes ($n = 2000$) in our scRNA-seq dataset. After that, we obtained the 9 nearest samples from the TCGA dataset for each cell according to their Spearman's correlation coefficients. The identity (malignant or non-malignant) for each thyrocyte was determined by the label which is most frequent among their nearest neighbors. To validate the efficiency of our classification, we utilized the PRJEB11591 dataset as the validation cohort and applied our algorithm to this bulk profile. It turned out that our method could well distinguish tumors and para-tumors with 97% sensitivity and 96% specificity, supporting the robustness of our malignant/non-malignant identification of thyrocytes in scRNA-seq data.

**Deconvolution analysis of the bulk RNA-seq profiles.** Deconvolution analysis of the integrated bulk RNA-seq data (combining both TCGA and PRJEB11591 profiles) against our scRNA-seq dataset was conducted using the BisqueRNA package with default settings[24]. We labeled our cells into 11 categories, including CD4$^+$ T cells, CD8$^+$ T cells, myeloid cells, B cells, NK cells, myoCAFs, iCAFs, ECs and three phenotypes of malignant thyrocytes found in trajectory analysis (follicular-like, p-EMT-like and dediff-like). The deconvoluted cell-type composition of each bulk sample was then utilized for group comparisons.

**Construction of myoCAF and iCAF scores for fibroblasts in the scRNA-seq dataset.** The myoCAFs and iCAFs are two distinct types of CAFs. Previous studies have identified their transcriptional distinctions in several cancer types, such as pancreatic ductal adenocarcinoma (PDAC) and bladder urothelial carcinoma[77,78]. To assess the potential transcriptional diversity of fibroblasts in PTCs, we constructed an myoCAF module and an iCAF module with the myoCAFs and iCAFs signature genes from the PDAC study[78]. Their module scores were computed by Seurat's AddModuleScore() function for each fibroblast. The detailed markers for myoCAFs and iCAFs were shown in Supplementary Table 11.

**Classification of ECs through label transfer method.** To characterize the heterogeneity of EC phenotypes in PTCs, we referred to another scRNA-seq dataset, which defined several types of ECs at single-cell resolution for lung cancer[34]. In detail, we first performed unsupervised clustering for ECs in our study. Next, the standard workflow of label transfer method was performed in Seurat by identifying the "anchors" between the two datasets and transferring the information into our dataset. In this way, the predicted phenotype for each EC in our scRNA-seq data was yielded.

**Cell-cell interaction analysis.** To investigate the potential interactions between different cell types in the TME of PTCs, we performed cell-cell interaction analysis using CellPhoneDB (python package, v2.1.4), which integrates a publicly available repository of curated ligand-receptor (L-R) pairs and a statistical framework[30]. As described above, our ~158,000 cells were grouped into 22 immune cells clusters, 2 fibroblasts clusters, 5 ECs clusters and malignant or non-malignant thyrocytes. Interaction networks between all these cell clusters were investigated. The $P$ value and average expression level for each L-R pair were obtained through the statistical framework of CellPhoneDB. The significant cell type-specific interactions between L-R pairs ($P < 0.05$, mainly cytokines, chemokines and growth factors) were selected for evaluation and visualization.

**SCENIC analysis.** Single-Cell rEgulatory Network Inference and Clustering (SCENIC) analysis was performed to reveal the gene regulatory network (GRN) in different cell types and clusters[79]. We performed the SCENIC analysis using the latest version of pySCENIC (v0.10.2), a lightning-fast python implementation of the SCENIC pipeline. The gene-motif rankings (500 bp upstream or 100 bp downstream of the transcription start site [TSS]) were used to determine the search space around the TSS. The motif database (mc9nr) including 24,453 motifs was used for RcisTarget and GENIE3 algorithms to infer the GRNs, respectively.

**Quantification and statistical analysis.** All the statistical analyses were performed using R (version 3.6.1). Student's $t$ test, Wilcoxon rank-sum test, Pearson's chi-square test, log-rank test, Pearson's correlation coefficient and Spearman's rank correlation coefficient were utilized in this study.

**Reporting summary.** Further information on research design is available in the Nature Research Reporting Summary linked to this article.

## Data availability

The processed scRNA-seq data generated in this study have been deposited in the Gene Expression Omnibus (GEO) database under accession code GSE184362. The raw scRNA-seq and whole-exome sequencing reads generated in this paper are deposited in Genome Sequence Archive (GSA) with the accession number HRA001107. Access to the

raw data may be requested by completing the application form via GSA-Human System and is granted by the corresponding Data Access Committee. The approximate response time for accession requests is about 10 working days. Additional guidance can be found at the GSA-Human website [https://ngdc.cncb.ac.cn/gsa-human/document/GSA-Human_Request_Guide_for_Users_us.pdf]. PTC and ATC scRNA-seq datasets (GSE158291 and GSE148673) were obtained from GEO for validation. Bulk RNA-sequencing dataset, including PRJEB11591 from Sequence Read Archive (SRA), and TCGA THCA datasets from Google Cloud Pilot RNA-Sequencing for CCLE and TCGA [https://osf.io/gqrz9/] were used in this study. Endothelial cell annotation was performed using the scRNA-seq dataset from the lung tumor endothelial cell taxonomy database [https://www.vibcancer.be/software-tools/lungTumor_ECTax]. The remaining data are available within the article, supplementary information and Source data. Source data are provided with this paper.

## Code availability

Experimental protocols and the data analysis pipelines used in our work followed the 10X Genomics and Seurat official websites. The analysis steps, functions and parameters used are described in detail in the Methods section. The scripts of key steps can be found at the GitHub repository: https://github.com/puweilin/scRNAseq_PTC.

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

## Acknowledgements

This study was supported by the grants from National Natural Science Foundation of China (81772851 and 81972501), the Basic Research Project from Technology Commission of Shanghai Municipality (17JC1402200), Young Scientists Fund of the National Natural Science Foundation of China (82003360, 82002830), Postdoctoral Science Foundation of China (2019M651354), Shanghai Sailing Program (20YF1408200) and Shanghai Municipal Science and Technology Major Project (2017SHZDZX01). We thank OE Biotech Co., Ltd (Shanghai, China) for providing whole exome sequencing, Dr. Dong An and Wu Wang for assistance with bioinformatics analysis for whole exome sequencing.

## Author contributions

Conceptualization, Y.-L.W., X.Z., J.W., Q.J., W.P., and X.S.; Resources, Y.-L.W., P.H., Y.W., D.J., H.G., W.W., Z.L., N.Q., H.L., Q.J., and X.Z.; Methodology, Y.-L.W., W.P., X.Z., and X.S.; Investigation, W.P., X.S., P.Y., M.Z., and L.T.; Formal Analysis, W.P., X.S., P.Y., X.Z., and Y.-L.W.; Validation, X.S., P.Y., M.Z., L.T., and J.H.; Data Curation, W.P., X.S., X.Z., and Y.-L.W.; Visualization, W.P., X.S., and P.Y.; Writing-Original Draft, X.S., Y.-L.W., W.P., X.Z., and P.Y.; Writing-Review & Editing, Y.-L.W., X.S., Z.L., X.Z., X.H., and W.P.; Supervision, Y.-L.W., X.Z., Y.W., J.W., and Q.J.

## Competing interests

The authors declare no competing interests.
