## [Peer review file · Nature Communications]

REVIEWER COMMENTS

Reviewer #2 (Remarks to the Author): Expert in thyroid cancer genetics, genomics, and therapy

Employing single cell sequencing Pu et al. analyzed 23 fresh samples (derived from 11 PTC patients) of diverse clinical course including paratumors, primary tumors, treated or recurrent lymph nodes, and RAIR distant metastasis. The authors evidently resolved thyrocytes and Tumor microenvironment (TME) dynamics in PTC's. The top transcriptional discriminants were extrapolated into TCGA and PRJEB1151 cohorts. The findings on BRAF-B-like sub-cluster is interesting and if consistent may have therapeutic implication and expression of TMSB4X along the course of PTCs is intriguing. Besides, the TDS and BRS status along the developmental trajectory of PTC (or disease progression) align with existing literature in the field (TCGA study & DOI: 10.1172/JCI85271); the insights on tumor immune deconvolution, characterization of fibroblasts & endothelial cells is interesting and in most part is consistent with what is known. Overall this manuscript presents a moderately enthusiastic finding, however with in-depth revision the data might be a resource to the field.

Specific comments:

1. Fig1 & Table S4: One of the major concerns is distribution of samples across the clusters, for example in Table S4 the thyrocyte cluster C03, C04 and C08 are predominately defined by cells from single sample i.e. LN7r, T9 and SC11 respectively, which questions inter-tumor heterogeneity? Additionally, contribution of thyrocytes from patient 2 (P2, LN2, T2) is completely absent (Table S4), is this a sample failure or other reason for exclusion ?
2. Fig 1f: There is a higher representation of CD4-C7-ISG15 subcluster in paratumor Vs tumors. Is there a known function of this CD4 subgroup that is worth discussing?
3. The authors must provide the genetic alterations (mutations and CNV) of all patients studied, the authors should consider annotating thyrocyte clusters with underlying genetic alterations and test whether accumulation of genetic alterations underlie disease progression/tumor microevolution and TME dynamics, should acknowledge if cannot be derived due to less sample number/low mutation burden. Furthermore, the projection of genetic alterations on paratumor clusters (c01 & c02) may improve clarity on authors classification of pre-malignant cells, if these clusters harbor genetic lesions at low allelic fractions which may have some effect at the transcriptional level but not completely transformed.
4. Fig3: The authors should cautiously approach in classifying c02 as pre-malignant cells that are primed for malignancy. Authors should discuss the various possibilities of these transcriptionally intermediate cell cluster (c02 Vs c01) and acknowledge weakness if any:
 - (a) Are these normal follicular cells that are in different phase of cell cycle or those that exit thyroid hormone biosynthesis? or cells that are modulated by the secretions from the adjacent tumor or TME?
 - (b) Does these cells harbor genetic lesion that has some effect at the transcriptional state but not fully transformed?
 - (c) 90% cells from this cluster are from a single patient (P5+T5; Table S4), how to justify this as a biologically recurring premalignant state?
5. Fig. 3c and Fig. S5e: It is important to add all 13 thyroid differentiation genes in these bubble plots. Alternatively, the authors could provide as a separate figure, the heat maps of all TDS, Braf & Ras signatures comparing (c01 vs c02 vs c3-c09 & state1 vs state2 vs state3).
6. Figure 3d: The H&E show patches of malignant cells, which doesn't seem to agree fully with authors claim of para-tumors as morphologically normal.
7. Figure 4b: Provide also a table in the supplementary for distribution of thyrocyte clusters (c01-c09)

within these 3 states of thyrocyte.

8. In relation to Fig. 5d: The prevailing concept in breast, melanoma etc. that MAPK pathway activation is inversely correlated to MHC-II expression/signaling (DOI: 10.1158/1078-0432.CCR-18-3200), but the data in this figure suggest that other way ?, if that is the case the authors should acknowledge the difference and discuss this relation in thyroid cancer.

9. Fig. 7j: The inclusion of single patient data pre and post treatment for combined famitinib and camrelizumab appears to be a stretch and it doesn't serve as a proof of concept. Reason 1: There is no evidence of the patient's transcriptional data that show this patient represents Braf-B-like tumor. Reason 2: There is no information on overall metrics of the trial data that show the number of responders, non-responders etc.

I would recommend taking this data out if authors are not ready to provide the inferential details of the trial justifying this as a proof of concept.

10. Fig. S10: It is highly speculative showing immune dynamics using a colon adenocarcinoma cells (MC-38) and extrapolate that to thyroid, the authors should show the evidence of immune dynamics with famitinib treatment using thyroid cancer cell lines or in patients.

11. Most of the figure labels are not legible. Font adjustments/reformatting is required if accepted for publication.

Reviewer #3 (Remarks to the Author): Expert in thyroid cancer genetics, therapy, mouse models, and angiogenesis

The authors have tried to prove tumor heterogeneity of papillary thyroid carcinoma (PTC).

Overall, this is a descriptive study without mechanistic insights. The translational applications of the descriptive results are not clear. More importantly, many experiments don't include positive and negative controls.

The study shows several limitations that decrease scientific enthusiasm:

1. I would start highlighting that in Discussion there is not comment about the limitations of the single cells analysis.
2. The study was designed using 11 patients. This is statistically a very low number to answer such important biological question (tumor diversity and progression).
3. Fig.1D: it represents important results. However, this analysis lacks of further analysis to characterize all cell lineages. E.g. the fibroblast population is highly heterogeneous. It is unknown yet all cell subtypes belonging to CAFs. These findings would had been a good opportunity to prove this. All markers in this figure are not validated. These are promiscuous markers.
4. Single cell analysis identified 11 categories, including CD4+ T cells, CD8+ T cells, myeloid cells, B cells, NK cells, myoCAFs, iCAFs, ECs and three phenotypes of malignant thyrocytes found in trajectory analysis (follicular-like, p-EMT-like and dediff-like): none of these cell types were validated in another independent cohort of PTC.
5. The cell type with dedifferentiation should be tested in anaplastic thyroid carcinoma cells, which represent the best model and status of thyroid tumor dedifferentiation.
6. All in vivo experiments were performed using a murine colon cancer cell line. Authors could isolate murine PTC cells from commercially deposited BRAFV600E mice that form PTC.
7. Lack of functional validation for iodine uptake and retention. It would be very helpful to validate the signatures and modulate their expression applying treatments with BRAFV600E inhibitors. This experiment might prove the dependency of the gene signatures identified through the BRAFV600E

pathway.

8. Thyroid differentiation score (TDS) is the same used in the TCGA (Cell 20214). The expectation was to see new gene profiles for thyroid differentiation in the 3 thyroid cell types. Instead, the standard TDS was used, without adding any new findings.

9. Fig5: BRAFV600E Ab and PDL1: there is not any statistical analysis of correlation or multivariate analysis of adjustment to confounding factors.

10. It would be helpful for readers to cite in the Discussion, previous findings on EMT in thyroid cancer: <https://pubmed.ncbi.nlm.nih.gov/21383698/>

11. Fig.7: this is an important design in the study. However, there is not any further validation in vitro and in vivo of these findings.

12. Fig.7J: it lacks of PET analysis, to prove the real metabolic status of the areas indicated from the arrows.

Reviewer #4 (Remarks to the Author): Expert in angiogenesis and endothelial cells

In this study, the authors aim to profile transcriptional diversity within papillary thyroid carcinoma (PTC) samples. They perform single cell RNA sequencing (scRNAseq) of PTC tumors and metastases in various stages from patients. Through their analyses, they found that PTC cells transition from a "cancer-primed" premalignant phenotype to three different malignant phenotypes, and one subtype is associated with worse prognosis. They also identified two types of cancer-associated fibroblasts within PTC tumors, and the inflammatory fibroblast phenotype has significant predicted interaction with immune cell types. Furthermore, they found that endothelial cells with the tip cell phenotype within tumors showed strong immune crosstalk. The authors conclude that combining anti-angiogenic therapy and immunotherapy may be beneficial in treating PTC. Although these studies are interesting and provide a large amount of clinical scRNA sequencing data to describe the transcriptional landscape of PTC tumors on a single cell level, there are several issues with data analysis and lack of validation to support conclusions, as detailed below.

Major Comments

1. The authors describe using functions within Seurat to account for batch effects between samples. Even with this processing, the clusters of thyrocytes within Figure 2 still seem to be largely described by the source tumor (Figure S2a). This would suggest that the variation within thyrocytes defining the different clusters is due to the patient rather than the cell type. This is a large problem with the analysis of the dataset that limits the strength of the conclusions on thyrocytes presented in Figures 2, 3, and 4. This means that over 75% of the transcriptional description of pre-malignant thyrocytes is provided by one sample source (and around 90% of the c01 cluster). The authors must add a detailed description of this limitation on the analysis within the text.

2. How does the pseudotime statistic (and pseudotime-associated genes) describe differentiation with the clear bifurcation presented in Figure 4? It looks like lower pseudotime values correlate to follicular-like cells, how are the authors distinguishing cells with greater pseudotime values as pEMT-like or dediff-like phenotype?

3. For the endothelial cell phenotype assignments, it is unclear what the "immature EC" phenotype is referencing. This cluster may be describing a stalk EC phenotype, or potentially containing ECs that are proliferating. Further analysis of the differential gene expression is necessary to determine this cluster.

4. The ligand/receptor relationships between tip cells and other cells are interesting. As the authors are claiming these interactions are potentially unique to tip cells, it would also be necessary to see the ligand/receptor relationships from venous and lymphatic ECs to other cell types to compare with tip cell interactions.

5. The results suggest that tip cells, that are pro-angiogenic, are also communicating with immune cells. Why would an anti-angiogenic therapy promote immune cell infiltration? Anti-angiogenic

therapies should inhibit tip cell phenotypes in the tumors, effectively reducing immune crosstalk. The results on vascular-immune crosstalk should be explained in more detail, along with describing the immune cell crosstalk with other endothelial cell phenotypes. Also, validation of the effects of famitinib and camrelizumab on vascular-immune crosstalk is necessary to conclude that these drugs may be working to inhibit PTC progression by blocking these interactions. For example, in vitro analysis of sprouting angiogenesis or co-culture with endothelial cells and immune cells could be analyzed to validate this hypothesized pathway.

Minor Comments

1. The authors highlight TMSB4X as a potential marker for malignancy by showing expression levels in c01, c02, and c03-c09 in Figure 3c. But this analysis does not consider single cells as they are transitioning. A more accurate visualization would be to show TMSB4X expression in single cells graphed on Component 1 versus Component 2, as shown in Figure 3f. Additionally, is there differential expression of TMSB4X within the malignant cell clusters shown in Figure 4?
2. Although the Methods section suggests that tSNE and UMAP dimensionality reduction analyses were performed when appropriate, it is not clear from the text why tSNE was shown in Figures 1, 3, 6, and 7, but UMAP was shown in Figure 2.
3. With the color choices, the subcutaneous population is difficult to see in Figure 1e (right panel).
4. More description of canonical markers used to identify different clusters as phenotypic populations would be helpful in all experiments.

Reviewer #5 (Remarks to the Author): Expert in single-cell RNA-seq and microenvironment analysis

In this manuscript, the authors generate a large-scale scRNA-seq dataset of papillary thyroid cancer (PTC) patients spanning a variety of disease states and metastatic sites. The authors perform extensive analysis of this dataset using trajectory analysis, cell communication to provide a comprehensive characterization of PTC.

This is a review of the technical aspects of single-cell RNA-seq analysis presented in the manuscript. The approaches are for the most part reasonable and technically sound but I request the authors address the following comments/concerns.

1. Trajectory analysis:

a. An important aspect to note is that any trajectory detection algorithm assumes that cells lie along a continuous manifold and when start cell is specified directionality is known. However, the authors use the application of trajectory detection as proof of existence of a continuous manifold - I think the authors should correct the interpretation here and demonstrate the existence of continuous manifolds using tools such as gene expression trends of known markers.

b. Are results presented in Figs. 3e-f reproducible across different patient samples?

2. The authors switch from Monocle to Diffusion maps for analysis of paraneoplastic thyrocytes to all thyrocytes - why was this switch necessary and why not use the same analysis approaches across the board? I am also surprised to see that the cluster structure is rather disrupted in Fig. 4b - in particular, c08 appears to be present in both branches. This indicates either the clustering or trajectory detection needs some correction since trajectory detection in general does not dramatically alter the relationships between cells.

3. Batch correction: In the first step, the authors apply batch correction (Fig. 1) - which inherently assumes all differences are technical. However, when zooming into thyrocytes the authors do not apply any batch correction (Fig. 2). The authors should clarify the assumptions and approach here - perhaps they could perform differential analysis between the thyrocyte patient samples and use gene set enrichment or similar approaches to examine whether differences are technical or if they represent

true biology.

4. In a couple of instances, the authors choose to highlight genes as being relevant to the biology of disease: (a) TMSB4X in Fig2 (b) PKHD1L1 in Fig. 3. What was the statistical criteria for these genes and are there genes at a similar level of significance? The authors better need to justify the selection of these genes given the weight of biological role attributed to them.

5. Classification of scRNA-seq data using bulk data: While the bulk data classifier really well, how do the authors take into account the drop-outs in scRNA-seq when transferring the model? It would be useful to examine the prediction scores to ensure dropouts do not adversely affect the single-cell annotation.

Responses to the Reviewers

Reviewer #2 (Remarks to the Author): Expert in thyroid cancer genetics, genomics, and therapy

Comment 1: *Fig1 & Table S4: One of the major concerns is distribution of samples across the clusters, for example in Table S4 the thyrocyte cluster C03, C05 and C09 are predominately defined by cells from single sample i.e., LN7r, T9 and SC11 respectively, which questions inter-tumor heterogeneity?*

Answer 1: Thanks very much for your question. We totally agree with you that inter-tumor heterogeneity is the primary reason for the distribution of samples across the clusters in our study for the following reasons.

First, our study performed MNN (mutual nearest neighbors) algorithm to correct the batch effect across samples, which is a commonly-used and efficient method¹. As shown in Figure 1B and Figure 1E, stromal cells and immune cells were integrated well after batch effect correction and showed little inter-sample heterogeneity, indicating that MNN has successfully removed batch effects in our study. Therefore, we suggested that the higher heterogeneity of the thyroid tumor epithelial cells (thyrocytes) between different samples was due to their biological differences, but not due to batch effects.

Second, to cover more tumor diversity, we collected samples from PTC patients with diverse clinical courses, bringing more inter-tumor heterogeneities. These inter-tumor heterogeneities may reflect the unique biological features of samples with different clinical courses and characteristics. □ Take SC11 as example, Patient 11 had a very advanced disease and received three times of radioactive iodine (RAI), a well-known reason that induces PTC dedifferentiation. Therefore, it is easy to understand why cells from her subcutaneous metastases (SC11) sample were predominantly enriched in the c09 cluster, which had the poorest differentiation level

(lowest TDS score). In other words, the predominant concentration of SC11 cells in the c09 cluster well reflects the dedifferentiated biological nature of SC11, and corresponds well with the clinical data. □ Take LN7r as another example, the right LN metastases (LN7r) of Patient 7 had prevalent and well-retained follicular architectures (Figure S6D), therefore it is easy to understand that LN7r was predominantly enriched in the c03 cluster, which had the highest differentiation level among the malignant thyrocyte clusters (c03-c09). In other words, the predominant concentration of LN7r cells in the c03 cluster well reflects the highly differentiated biological nature of LN7r, and corresponds well with the clinical data.

Third, to further clarify that the differences between these clusters reflected the inter-tumor heterogeneity or potential technical noises, we performed pathway enrichment for the c03, c05, c06 and c09 clusters, which are mainly defined by cells from a single sample. The results showed that these clusters showed distinct enriched functional pathways from the others (**Table S7**), suggesting their differences are more likely to be biological diversities of the samples.

Taken together, we think the distribution of samples across the clusters may be largely determined by the inter-tumor heterogeneity, but not batch effects between our samples. These clinical and biological diversities between samples could provide many important findings in our study.

Thank you very much. We hope you will be satisfied with our response.

Comment 2: *Additionally, contribution of thyrocytes from patient 2 (P2, LN2, T2) is completely absent (Table S4), is this a sample failure or other reason for exclusion ?*

Answer 2: Thanks for the reviewer's question. In our study, we have adopted a standard and consistent sample digestion and preparation procedure for each sample and each tissue, as described in the Methods section. We did not perform any exclusion procedures to these samples.

As for Patient 2, we obtained P2, LN2 and T2 simultaneously and digested these tissues as usual. We suggest that the absence of thyrocytes in these samples might be

caused by enzyme activity or other unknown issues in the preparation of single-cell suspension. Despite this issue, we believe that other cell types from Patient 2, such as immune cells, still have an important contribution to the cell pool of our study. Therefore, Patient 2 is still included in our study.

Thank you very much. We hope you will be satisfied with our response.

Comment 3: *Fig 1f: There is a higher representation of CD4-C7-ISG15 subcluster in paratumor vs tumors. Is there a known function of this CD4 subgroup that is worth discussing?*

Answer 3: Thanks for your advice. According to your suggestion, we have performed a literature review and found that *ISG15* plays a vital role in innate immune response ². Extracellular *ISG15* could function as a cytokine with several immunomodulatory activities, including the induction of natural killer (NK) cell proliferation ³ and dendritic cell maturation ⁴. However, the functions of the subcluster in PTC remain largely unknown.

In the Revised Manuscript, we have added a brief discussion in the Results section as follows.

“Furthermore, the ISG15-expressing CD4+ T cells were significantly enriched in paratumors. Previous reports suggested that ISG15 might participate in natural killer (NK) cell proliferation, dendritic cell maturation, or other innate immune responses ^{2, 3, 4}. However, the definite role of this subcluster in PTC needs to be further examined.”

Thank you very much. We hope you will be satisfied with our response.

Comment 4: *The authors must provide the genetic alterations (mutations and CNV) of all patients studied, the authors should consider annotating thyrocyte clusters with underlying genetic alterations and test whether accumulation of genetic alterations underlie disease progression/tumor microevolution and TME dynamics, should acknowledge if cannot be derived due to less sample number/low mutation burden.*

Furthermore, the projection of genetic alterations on paratumor clusters (c01 & c02) may improve clarity on authors classification of pre-malignant cells, if these clusters harbor genetic lesions at low allelic fractions which may have some effect at the transcriptional level but not completely transformed.

Answer 4: Thanks for the reviewer's comment and suggestion. In full accordance with your advice, we performed Whole-Exome Sequencing (WES) plus Sanger sequencing for all patients studied. It is noteworthy that DNA sequencing is not included in our original study design, therefore fresh samples including germline reference samples (e.g., blood, saliva) were not collected and stored at that time. Consequently, our DNA sequencing analysis followed a tumor-only pattern with only formalin-fixed, paraffin-embedded (FFPE) tumor tissues available. Following your suggestion, we have summarized the key driver mutations for each patient in Table S2 and S3 of the Revised Manuscript. In addition, the raw data has been submitted to GSA (Genome Sequence Archive, <https://bigd.big.ac.cn/gsa/>) with the accession number PRJCA005863.

We totally agree with your suggestion that it would be important to annotate these clusters using genetic alterations, particularly for the premalignant thyrocytes (c02). However, the WES or Sanger sequencing is performed and analyzed at the bulk-sample resolution, reflecting the genomic status of bulk samples with mixed cell types, and their results could not be directly used for thyrocyte annotation, which is at the single-cell resolution. We are afraid that using bulk-sample-level genomic data to annotate single-cell level cell clusters is currently unachievable.

We consider your suggestion of great importance and have added a detailed limitation in the Revised Manuscript as below.

“Third, our study only examined the mutational status of bulk samples, while the genetic annotation of different thyrocyte clusters is lacking. Future studies with integration of multimodal single-cell sequencing data may delineate the DNA mutation spectrum, epigenetic modifications and transcriptome at the single-cell resolution simultaneously for different cell clusters.”

Thank you very much. We hope you will be satisfied with our response.

Comment 5: *Fig3: The authors should cautiously approach in classifying c02 as pre-malignant cells that are primed for malignancy. Authors should discuss the various possibilities of these transcriptionally intermediate cell cluster (c02 vs c01) and acknowledge weakness if any:*

(a) Are these normal follicular cells that are in different phase of cell cycle or those that exit thyroid hormone biosynthesis? or cells that are modulated by the secretions from the adjacent tumor or TME?

(b) Do these cells harbor genetic lesion that has some effect at the transcriptional state but not fully transformed?

(c) 90% cells from this cluster are from a single patient (P5+T5; Table S4), how to justify this as a biologically recurring premalignant state?

Answer 5: Thanks for the reviewer's suggestion. In full accordance with your comment, we provided detailed explanations to each concern as follows.

First, for your concern (a), we assessed the cell cycles of the cells from c01 and c02 using the "CellCycleScoring" function in Seurat. The cell cycles of these cells were shown below and we found no significant differences in cell cycles between c01 and c02 thyrocytes, suggesting that different cell cycle phases may not induce the differences.

Table. The percentage of c01 and c02 cells in different cell cycle states

Cluster	G1 (%)	S (%)	G2M (%)
c01	44	15	40
c02	48	10	42

Second, for your concern (a), in the Revised Manuscript, we have added **Figure S2C** (as shown below), suggesting that the c02 premalignant cells showed higher

level expression of TDS-associated genes than the malignant cells, indicating that they have not exited thyroid hormone biosynthesis.

Third, for your concern (a), we totally agree with your suggestion that the premalignant cells are modulated by the secretions from the TME. However, TME consists of multiple cells, including immune cells, stromal cells and adjacent thyrocytes, and their effects on the premalignant cells may not be quantified and compared precisely. Therefore, it is a great pity that we could not provide a clear conclusion to determine if the differences between c01 and c02 were caused by modulation from adjacent tumor or TME.

Fourth, for your concern (b), as described above in Answer 4, we totally agree with your comment that these premalignant cells may harbor genetic lesions that affected the transcriptional state but are not fully transformed to malignant cells. However, to validate it, we should simultaneously perform single-cell DNA sequencing (scDNA-seq) together with scRNA-seq. Therefore, we are sorry for the fact that we could not identify the potential genetic alterations and their effects at the

transcriptional state for c02 cells with the scRNA-sequencing data.

Fifth, for your concern (c), “pre-malignant cells” in the c02 cluster could only be defined when simultaneously meeting the following three aspects. (1) histologically normal follicular architecture (outwardly normal); (2) different from c01 (normal thyrocytes) in the evolutionary paths; (3) the most important, the potency of multifocal independent tumorigenesis confirmed by pathologists. This kind of samples, especially those meeting the criteria (3), are uncommon in clinical practice. Due to the limited number of samples in our study (23 samples from 11 patients), we only found the pre-malignant cells in one patient (Patient 5), which is still a surprise and highlight. We totally agree with you that we should perform studies with a larger sample size to confirm the pre-malignant state of thyrocyte as biologically recurring.

In summary, for the three points above, just as your suggestion, we must admit that the pre-malignant cells in our study need to be further verified. Therefore, in full accordance with your comment, we have acknowledged the weakness of the findings for the pre-malignant cells in the Revised Manuscript as follows.

“We acknowledge that our study has several limitations. First, the sample size and cell number in our study is limited, and further expansion of the study cohort is required to better elucidate the tumor progression and diversity in PTC. In particular, the pre-malignant cells found in our study are primarily determined by one patient, and this state of thyrocytes should be verified in further studies.”

“Third, our study only examined the mutational status of bulk samples, while the genetic annotation of different thyrocyte clusters is lacking. Future studies with integration of multimodal single-cell sequencing data may delineate the DNA mutation spectrum, epigenetic modifications and transcriptome at the single-cell resolution simultaneously for different cell clusters.”

“Although the universality of the pre-malignant thyrocytes needs to be further validated in future studies, these observations and hypotheses, to some extent at least, provide a mechanistic insight into the multicenter onset of tumorigenesis in a fraction of PTCs.”

Thank you very much. We hope you will be satisfied with our response.

Comment 6: *Fig. 3c and Fig. S5e: It is important to add all 13 thyroid differentiation genes in these bubble plots. Alternatively, the authors could provide as a separate figure, the heat maps of all TDS, Braf & Ras signatures comparing (c01 vs c02 vs c3-c09 & state1 vs state2 vs state3).*

Answer 6: Thanks for the reviewer's suggestion. In full accordance with your advice, in the Revised Manuscript, we have provided the heatmaps of all TDS, *BRAF* and *RAS* signatures between different clusters and states in *Figure S2c* and *Figure S5g-h*.

Comment 7: *Figure 3d: The H&E show patches of malignant cells, which doesn't seem to agree fully with authors claim of para-tumors as morphologically normal.*

Answer 7: Thanks for the reviewer's comment. We totally respect your suggestion that the H&E figure does not fully agree with our claim of para-tumors as morphologically normal. We will provide a detailed answer to this concern.

First, our study follows a standard procedure of sample collection, the para-tumor samples were defined with the following criteria:

- 1) More than 1cm away from the visible tumor foci;
- 2) Tumor-free in ultrasound and CT scan prior to surgery;
- 3) Tumor-free by naked eye observation during intraoperative sample collection;
- 4) The fresh samples could be immediately dissolved into single-cell suspensions to maintain cell activities.

The P5 sample showed no apparent abnormalities in preoperative ultrasound, CT scan, and intraoperative observation by naked eyes, which completely followed the abovementioned criteria. As you can see in Figure 3D, these patches of tumor cells are very small that are only visible under the microscope. In other words, during the process from sample collection to sample preparation to single-cell sequencing, the P5 sample meets all of the criteria of the para-tumor definition. Therefore, we suggested that the paratumors of P5 could be claimed as morphologically normal.

Second, the specificity of P5 sample is accidental, but provides important clues to the PTC tumorigenesis. Despite some limitations described in Answer 5, the specificity of the P5 sample provided mechanistic insight into the multicenter onset of tumorigenesis in a fraction of PTCs. Moreover, we have confirmed that the small patches of tumor cells in P5 are not due to contamination or dissemination of tumor cells in T5, and they are actually more likely to derive from premalignant thyrocytes (the surrounding normal-appearing cells) in P5.

Thank you very much. We hope you will be satisfied with our response.

Comment 8: *Figure 4b: Provide also a table in the supplementary for distribution of thyrocyte clusters (c01-c09) within these 3 states of thyrocyte.*

Answer 8: Thanks for the reviewer's suggestion. In full accordance with your advice, we have provided the table summarizing the distribution of thyrocyte clusters within different cell states as Table S13.

Comment 9: *In relation to Fig. 5d: The prevailing concept in breast, melanoma etc. that MAPK pathway activation is inversely correlated to MHC-II expression/signaling (DOI: 10.1158/1078-0432.CCR-18-3200), but the data in this figure suggest that other way? if that is the case the authors should acknowledge the difference and discuss this relation in thyroid cancer.*

Answer 9: Thanks for the reviewer's question. Your advice is really helpful and constructive. We totally agree with you that the correlation between MAPK pathway activation and MHC-II signaling is important and should be examined in our study. In full accordance with your comment, we have quantified the transcriptional MAPK pathway activation level between *BRAF*-like-A and *BRAF*-like-B groups using the MAPK Pathway Activity Score (MPAS) according to a previous study ⁵.

As the figure showed below, we found that the MAPK score of the *BRAF*-like-B group was significantly lower than that of the *BRAF*-like-A group. In accordance, Fig.

5d showed the *BRAF*-like-B samples had a higher pathway activation of MHC-II expression/signaling. Taken together, these results in our study are in full accordance with the study (DOI: 10.1158/1078-0432.CCR-18-3200) you mentioned in this comment.

Thank you very much. We hope you will be satisfied with our response.

Comment 10: *Fig. 7j: The inclusion of single patient data pre and post treatment for combined famitinib and camrelizumab appears to be a stretch and it doesn't serve as a proof of concept. Reason 1: There is no evidence of the patient's transcriptional data that show this patient represents BRAF-like-B tumor. Reason 2: There is no information on overall metrics of the trial data that show the number of responders, non-responders etc. I would recommend taking this data out if authors are not ready to provide the inferential details of the trial justifying this as a proof of concept.*

Answer 10: Thanks for the reviewer's suggestion. We fully appreciate your suggestion and admit that the inclusion of the single patient data could not provide enough support for our conclusion. Meanwhile, we are afraid that we could not provide the details of the clinical trial due to the data privacy agreement. In addition, the Editor also suggested us to delete this clinical trial data because this trial has not been published. Therefore, in full accordance with both your and the Editor's advice, we removed the clinical trial data shown in Figure 7J and its related discussion in the Revised Manuscript.

Thank you very much. We hope you will be satisfied with our response.

Comment 11: *Fig. S10: It is highly speculative showing immune dynamics using a colon adenocarcinoma cells (MC-38) and extrapolate that to thyroid, the authors should show the evidence of immune dynamics with famitinib treatment using thyroid cancer cell lines or in patients.*

Answer 11: Thanks for the reviewer's suggestion. Both the Editor and other reviewers suggested us to remove the famitinib experiment data in the Revised Manuscript due to the following reasons, and we followed their advice.

First, the reason that we used murine colon cancer cell line as an alternative is totally due to the fact that there is still a lack of commercially available murine (mouse-derived) papillary thyroid cancer cell line. We admit that it is not appropriate to use a murine colon cancer cell line in this thyroid cancer study.

Second, due to the data privacy policy of unpublished clinical trial data, both the Editor and you suggested us to remove the famitinib clinical trial data, as mentioned in Answer 10. As famitinib clinical trial data were removed, the famitinib animal experiment should also be removed.

Due to the reasons above, we removed these experiments in our Revised Manuscript following the Editor's and other reviewers' advice.

Moreover, due to the lack of *in vivo* and *in vitro* experiments in our study, we have acknowledged the limitations of our study in the Discussion section as follows.

“At last, our study performed a series of bioinformatic tools to dissect the TME of PTC, while the functional validation of these findings is lacking, such as the iodine uptake and retention in different kinds of thyrocytes, and the key genes in PTC tumorigenesis. Further in vivo and in vitro experiments are awaited to validate these findings in the future.”

Thank you very much. We hope you will be satisfied with our response.

Comment 12: *Most of the figure labels are not legible. Font adjustments/reformatting*

is required if accepted for publication.

Answer 12: Thanks for the reviewer's suggestion. In full accordance with your comment, we have reformatted the figure labels of all Figures and Supplementary Figures of the Revised Manuscript. We hope you will be satisfied with our revised figures. Thank you.

Reviewer #3 (Remarks to the Author): Expert in thyroid cancer genetics, therapy, mouse models, and angiogenesis

Comment 1: *I would start highlighting that in Discussion there is not comment about the limitations of the single-cell analysis.*

Answer 1: Thanks for the reviewer's comment. We totally agree with you that describing the limitations of our single-cell analysis is necessary. In full accordance with your comment, we have added the discussion of the limitations of our study as follows:

“We acknowledge that our study has several limitations. First, the sample size and cell number in our study is limited, and further expansion of the study cohort is required to better elucidate the tumor progression and diversity in PTC. In particular, the premalignant cells found in our study are primarily determined by one patient, and this state of thyrocytes should be verified in further studies. Second, current scRNA-seq must be performed within a short period of time after sample acquisition because fresh samples with viable cells are needed in this technology. It is still a challenge to completely correct the batch effects while fully retaining the true biological properties caused by inter-tumor heterogeneities. Though we performed the MNN algorithm for batch effect correction, batch effects may still exist in our study. Third, our study only examined the mutational status of bulk samples, while the genetic annotation of different thyrocyte clusters is lacking. Future studies with integration of multimodal single-cell sequencing data may delineate the DNA mutation spectrum, epigenetic modifications and transcriptome at the single-cell resolution simultaneously for different cell clusters. At last, our study performed a series of bioinformatic tools to dissect the TME of PTC, while the functional validation of these findings is lacking, such as the iodine uptake and retention in different kinds of thyrocytes, and the key genes in PTC tumorigenesis. Further in vivo and in vitro experiments are awaited to validate these findings in the future.”

Thank you very much. We hope you will be satisfied with our response.

Comment 2: *The study was designed using 11 patients. This is statistically a very low number to answer such important biological question (tumor diversity and progression).*

Answer 2: Thanks for the reviewer's comment. We totally agree with your comment that tumor diversity and progression are important biological questions. For a long time, we did not have effective methods to answer this question. Recently, the fast development of single-cell sequencing technology has provided us an excellent tool to dissect the tumor diversity and progression at the single-cell resolution. Therefore, we implemented scRNA-sequencing in our study to explore this vital problem in PTC. To answer the tumor diversity and progression requires us to collect samples with diverse clinical characteristics, in our study, we tried our best to cover the PTC diversity as much as possible by collecting samples from PTC patients with diverse clinical courses, including paratumor, localized/advanced tumors, initially-treated/recurrent LNs and RAI-resistant distant metastases. However, an important limitation of scRNA-sequencing is that fresh samples with viable cells are needed, which limited the number of samples included in a single study.

Just as your suggestion, we admit that the sample size in our study is still limited to answer such important biological question (tumor diversity and progression), and we have added the description of our limitations in the Revised Manuscript as follows:

“We acknowledge that our study has several limitations. First, the sample size and cell number in our study is limited, and further expansion of the study cohort is required to better elucidate the tumor progression and diversity in PTC.”

Thank you very much. We hope you will be satisfied with our response.

Comment 3: *Fig.1D: it represents important results. However, this analysis lacks of further analysis to characterize all cell lineages. E.g., the fibroblast population is highly heterogeneous. It is unknown yet all cell subtypes belonging to CAFs. These*

findings would had been a good opportunity to prove this.

Answer 3: Thanks for the reviewer's comment. In full accordance with your comment, we analyzed more canonical biomarkers for the fibroblast population.

First, previous studies have suggested several biomarkers for cancer-associated fibroblasts (CAFs), including α -smooth muscle actin (α -SMA), S100A4, PDGF- α , and vimentin (VIM) ^{6,7}. Therefore, we have checked the expression levels of these CAF biomarkers in our dataset. As shown below, we found that all fibroblasts in our dataset expressed at least one of these three CAF biomarkers. Therefore, we suggested that all fibroblasts in our study belong to CAFs regardless of their tissue types.

Second, we have added the description of the CAFs in the Revised Manuscript as follows:

“In our scRNA-seq dataset, we found that all fibroblasts, regardless of tissue types, expressed the canonical CAF biomarkers, including VIM, S100A4, ACTA2 (α -SMA), and PDGFRA, and were confidently defined as CAFs (Figures S9A-S9B).”

Thank you very much. We hope you will be satisfied with our response.

Comment 4: *All markers in this figure are not validated. These are promiscuous markers.*

Answer 4: Thanks for the reviewer's comment. We make sure that we used canonical markers for cell cluster classification in the previous manuscript. But we feel sorry

that in the previous submission, we did not annotate these clusters with canonical markers, instead we only annotated the most significantly differentially expressed genes (DEGs) in the previous submission (Figure 1D). Thank you again for this comment, now in this Revised Manuscript, we have revised the markers for each cell population them in Figure 1D.

In our study, all cells were classified into six major cell populations based on canonical markers, including T/NK cells (*CD3D*, *CD3E*, *CD3G*, *CD247*), B cells (*CD79A*, *CD79B*, *IGHM*, *IGHD*), thyrocytes (*TG*, *EPCAM*, *KRT18*, *KRT19*), myeloid cells (*LYZ*, *S100A8*, *S100A9*, *CD14*), fibroblasts (*ACTA2* [α -SMA], *COL1A1*, *COL1A2*, *COL3A1*) and endothelial cells (*PECAM1*, *CD34*, *CDH5*, *VWF*). All these markers have been commonly utilized for cell type identification in multiple previous single-cell sequencing studies^{8, 9, 10, 11, 12, 13}.

In the Revised Manuscript, we have revised the markers in the new Figure 1D, which included the canonical markers that are differentially expressed for each cell type. Meanwhile, we have changed the description of these cell types in the Revised Manuscript as follow:

*“After integrating the transcriptional data from all acquired cells, we primarily applied low-resolution t-distributed stochastic neighbor embedding (t-SNE) clustering and identified six main cell populations, which were labeled as T/natural killer (NK) cells (*CD3D*, *CD3E*, *CD3G*, *CD247*), B cells (*CD79A*, *CD79B*, *IGHM*, *IGHD*), thyrocytes (*TG*, *EPCAM*, *KRT18*, *KRT19*), myeloid cells (*LYZ*, *S100A8*, *S100A9*, *CD14*), fibroblasts (*COL1A1*, *COL1A2*, *COL3A1*, *ACTA2*) and endothelial cells (*PECAM1*, *CD34*, *CDH5*, *VWF*) (Figures 1C, 1D and S1C).”*

Thank you very much. We hope you will be satisfied with our response.

Comment 5: *Single cell analysis identified 11 categories, including CD4+ T cells, CD8+ T cells, myeloid cells, B cells, NK cells, myoCAFs, iCAFs, ECs and three phenotypes of malignant thyrocytes found in trajectory analysis (follicular-like, p-EMT-like and dediff-like): none of these cell types were validated in another independent cohort of PTC.*

Answer 5: Thanks for the reviewer's comment. In full accordance with your comment, we validated these cell types in another two independent cohorts of PTC totally following your advice.

To validate the cell type categories found in our study, we performed both literature review and scRNA-sequencing data analysis in another independent study. In summary, both two methods verified the existence of these cell types in PTC.

- 1) **Literature review:** We have carefully searched all the PTC single-cell sequencing studies and found two studies. In the preprint website MedRxiv, we searched a PTC scRNA-seq study performed by Wang Z et al ¹⁴. In that study, they have found 10 major cell types of PTC, which were mostly concordant with the cell types in our study, including **CD4⁺ T cells** (*CD3D+CD4+*); **CD8⁺T cells** (*CD3D+CD8A+*); **naive T cells** (*CD3D+CCR7+*); **B cells** (*CD19+MS4A1+CD38+CD79A+CD79B+*); **myeloid cells** (*CD14+CD86+ITGAX+CD80+CD83+ITGAM+*); **NKT cells** (*CD3D+NKG7+*); **endothelial cells** (*CD31+CD34+*); **epithelial cells** (*EPCAM+KRT18+*); **plasma cells** (*CD79A+SDC1+*); and **fibroblasts** (*COL1A1+*).
- 2) **scRNA-sequencing data analysis:** Peng et al. have performed a single-cell sequencing study to reveal the differences between male and female PTC patients using two pooled samples ¹⁵. We downloaded their raw data and performed quality control procedures, and processed their dataset using the same methods as those in our study. In total, we generated 10369 cells for further analysis. To verify the existence of the 11 cell types in this independent PTC cohort, we performed the standard workflow of the label transfer method built-in Seurat by identifying the “anchors” between the two datasets and transferring the information into their dataset. As shown below, we found that all the 11 categories defined in our study could be found in this dataset, suggesting that our classification of the cell types is robust and reproducible.
- 3) In full accordance with your comment, we have added the validation results (figure listed below) using Peng et al.'s data as the new Figures S1F and S1G and

added a description in our Revised Manuscript as follows:

“In addition, all these cell types were further validated using another scRNA-seq study of PTC (Figures S1F and S1G).”

Thank you very much. We hope you will be satisfied with our response.

Comment 6: *The cell type with dedifferentiation should be tested in anaplastic thyroid carcinoma cells, which represent the best model and status of thyroid tumor dedifferentiation.*

Answer 6: Thanks for the reviewer’s suggestion. In full accordance with your comment, we have assessed the correlation between the cell type with dedifferentiation (State 3 thyrocytes) and anaplastic thyroid carcinoma (ATC) cells, which successfully validated our results.

We have downloaded the scRNA-seq data of ATC cells from the GEO database (GSE148673). In total, 5586 ATC tumor cells were obtained and were further compared with our PTC thyrocytes. Using all shared genes between PTC and ATC single cells, we found that the correlation between our State 3 thyrocytes and ATC cells was as high as 0.72.

In full accordance with your comment, we added the sentence as follows: *“Furthermore, we obtained the scRNA-seq data of anaplastic thyroid carcinoma (ATC)*

from the Gene Expression Omnibus (GEO, accession number: GSE148673) database and analyzed all the shared genes with our PTC profiles. We found that the Pearson's correlation between State 3 thyrocytes and ATC cells was as high as 0.72.”.

Thank you very much. We hope you will be satisfied with our response.

Comment 7: *All in vivo experiments were performed using a murine colon cancer cell line. Authors could isolate murine PTC cells from commercially deposited BRAF^{V600E} mice that form PTC.*

Answer 7: Thanks for the reviewer's suggestion. Both the Editor and other reviewers strongly suggested us to remove all the famitinib *in vivo* experiment data in the Revised Manuscript due to the following reasons, and we followed their advice.

First, the reason that we used murine colon cancer cell line as an alternative is totally due to the fact that there is still a lack of commercially available murine (mouse-derived) papillary thyroid cancer cell line. We admit that it is not appropriate to use a murine colon cancer cell line in this thyroid cancer study.

Second, due to the data privacy policy of unpublished clinical trial data, both the Editor and reviewer suggested us to remove the famitinib clinical trial data. As famitinib clinical trial data were removed, the famitinib animal experiment should also be removed.

Due to the reasons above, we removed these experiments in our Revised Manuscript following the Editor's and other reviewers' advice.

Moreover, due to the lack of *in vivo* and *in vitro* experiments in our study, we have acknowledged the limitations of our study in Discussion as follows.

“At last, our study performed a series of bioinformatic tools to dissect the TME of PTC, while the functional validation of these findings is lacking, such as the iodine uptake and retention in different kinds of thyrocytes, and the key genes in PTC tumorigenesis. Further in vivo and in vitro experiments are awaited to validate these findings in the future.”.

Thank you very much. We hope you will be satisfied with our response.

Comment 8: *Lack of functional validation for iodine uptake and retention. It would be very helpful to validate the signatures and modulate their expression applying treatments with $BRAF^{V600E}$ inhibitors. This experiment might prove the dependency of the gene signatures identified through the $BRAF^{V600E}$ pathway.*

Answer 8: Thanks for the reviewer's suggestion. We totally agree with you that iodine uptake and retention after $BRAF^{V600E}$ inhibitor is important and should be addressed. Therefore, as suggested by the Editor (see Editor comment 4), we performed both literature review and transcriptome data analysis to assess the expression level of the TDS genes after treatment of $BRAF^{V600E}$ inhibitors. The results are shown as below.

- 1) **Literature review:** We reviewed the previous literature and found a prior study has found that the BRAF inhibitor vemurafenib restores RAI uptake and efficacy in a subset of $BRAF$ -mutant RAI-resistant (RAIR) patients, probably by up-regulating thyroid-specific gene expression via MAPK pathway inhibition¹⁶.
- 2) **Transcriptome data analysis:** Moreover, we found a public transcriptome dataset in the GEO database, which detected the transcriptome kinetics of SW1736 anaplastic thyroid cancer cell line at various times after the addition of 2 μ M vemurafenib (GSE37441)¹⁷. Therefore, we analyzed the transcriptome changes of the TDS genes after vemurafenib treatment. As shown in the figure below, we found that the majority of TDS genes were up-regulated after vemurafenib treatment for 6h or 48h.

Therefore, based on these results, we suggested that the $BRAF^{V600E}$ inhibitor could up-regulate and restore the expression level of TDS genes in PTCs. However, due to the lack of *in vivo* and *in vitro* experiments in our study, as suggested by the Editor, we have acknowledged the concern by describing a limitation as follows.

“At last, our study performed a series of bioinformatic tools to dissect the TME of PTC, while the functional validation of these findings is lacking, such as the iodine uptake and retention in different kinds of thyrocytes, and the key genes in PTC

tumorigenesis. Further *in vivo* and *in vitro* experiments are awaited to validate these findings in the future.”.

Thank you very much. We hope you will be satisfied with our response.

Comment 9: *Thyroid differentiation score (TDS) is the same used in the TCGA (Cell 2014). The expectation was to see new gene profiles for thyroid differentiation in the 3 thyroid cell types. Instead, the standard TDS was used, without adding any new findings.*

Answer 9: Thanks for the reviewer’s comment. This is a valuable suggestion, and we believed that your suggestion would be vital to improve the quality of our study. In full accordance with your comment, in the Revised Manuscript, we added new findings by identifying novel thyroid differentiation-related genes using our scRNA-seq dataset. The revision was performed as follows.

We first performed correlation analysis between gene expression and standard TDS score in our study. In total, we found 22 genes to be positively associated with the TDS score (Pearson’s $R > 0.5$) in our dataset (Table S10). Nine genes (*TG*, *TPO*, *TFF3*, *SLC26A4*, *DIO2*, *PAX8*, *TSHR*, *FOXE1* and *FHL1*) were previously used to

construct the standard TDS score. Moreover, the remaining 13 genes (*SLC26A7*, *IYD*, *MT1F*, *SORBS2*, *MT1G*, *HSP90B1*, *SORD*, *SLC26A4-AS1*, *CPQ*, *PRDX1*, *PKHD1L1*, *FCGBP*, *MATN2*) were first found to be associated with TDS score at the single-cell resolution. Among these 13 genes, *IYD*, *SLC26A7*, *HSP90B1*, *CPQ*, *PKHD1L1* have been previously reported to be involved in thyroid differentiation^{18, 19, 20, 21}. However, the remaining eight genes were newly identified candidate genes involved in thyroid differentiation, and the functions of these genes and their roles in PTC deserve further exploration.

In the Revised Manuscript, we have added the description of these candidate genes associated with thyroid differentiation as follows:

*“In addition, we identified 22 genes to be significantly and positively associated with TDS score (Pearson’s $R > 0.5$, $P < 0.05$) at single-cell resolution (Table S10). Among them, eight genes (*MT1F*, *SORBS2*, *MT1G*, *SORD*, *SLC26A4-AS1*, *PRDX1*, *FCGBP*, *MATN2*) have not been reported involved in thyroid differentiation. The functions and roles of these genes in PTC tumorigenesis deserve further exploration.”.*

Thank you very much. We hope you will be satisfied with our response.

Comment 10: *Fig5: BRAF^{V600E} Ab and PDL1: there is not any statistical analysis of correlation or multivariate analysis of adjustment to confounding factors.*

Answer 10: Thanks for the reviewer’s comment. We totally understand and agree with your concern, but we are sorry for the lack of statistical analysis in Figure 5E, which showed the immunohistochemistry (IHC) staining (using BRAF^{V600E} and PD-L1 antibodies) for Patient 10.

In our study, we found that *BRAF*-like-B tumors showed a higher activation level of PD-1 signaling than that in *BRAF*-like-A tumors through GSEA analysis (Figure 5D). In accordance, Patient 10 was used as an example (or a representative case) showing an advanced primary tumor (a typical *BRAF*-like-B tumor) with prominent immune infiltration, positive expression of *BRAF*^{V600E}-mutant protein and programmed death-ligand 1 (PD-L1) protein, which supported the characteristics of the

BRAF-like-B subgroup. As Patient 10's IHC staining is used only as an example or in other words, a representative case, therefore multivariate analysis was not performed.

We acknowledge that the samples in our study are limited and we could not perform the statistical analysis to further verify this finding. Therefore, we have added a short description of limitation regarding Figure 5E of the Revised Manuscript as follows (red text below).

“In our scRNA-seq cohort, Case 10 represented a typical example of BRAF-like-B subtype who demonstrated an advanced primary tumor with prominent immune infiltration, positive expression of BRAF^{V600E}-mutant protein and programmed death-ligand 1 (PD-L1) protein (Figure 5E), and further studies with more BRAF-like-B samples are required for further verification.”

Thank you very much. We hope you will be satisfied with our response.

Comment 11: *It would be helpful for readers to cite in the Discussion, previous findings on EMT in thyroid cancer: <https://pubmed.ncbi.nlm.nih.gov/21383698/>*

Answer 11: Thanks for the suggestion. We totally agreed that this study is important for EMT in PTC. In full accordance with your advice, we added this citation in the Discussion section together with several sentences for description as follows: *“Meanwhile, Knauf et al. found that PTC could further develop into poorly-differentiated thyroid cancer (PDTC) through MAPK-dependent EMT process. This important finding is partly consistent with the development of p-EMT/dediff-like thyrocytes in our study, as KRAS signaling of State 2/3 thyrocytes was activated, which in turn activated the MAPK pathway⁵².”* Thank you.

Comment 12: *Fig.7: this is an important design in the study. However, there is not any further validation in vitro and in vivo of these findings.*

Answer 12: Thanks for the reviewer's suggestion. As suggested by the Editor and other reviewers, we have removed our in vivo experiments in the Revised Manuscript.

the reasons were explained above in Answer 7.

We admitted that the lack of *in vivo* and *in vitro* validation is a major limitation in our study, and in accordance with your comment, we added a detailed description of the limitations in the Revised Manuscript as follows:

“At last, our study performed a series of bioinformatic tools to dissect the TME of PTC, while the functional validation of these findings is lacking, such as the iodine uptake and retention in different kinds of thyrocytes, and the key genes in PTC tumorigenesis. Further in vivo and in vitro experiments are awaited to validate these findings in the future.”

Thank you very much. We hope you will be satisfied with our response.

Comment 13: *Fig.7J: it lacks of PET analysis, to prove the real metabolic status of the areas indicated from the arrows.*

Answer 13: Thanks for the reviewer’s suggestion. Both the Editor and other reviewers suggested us to remove the clinical trial data (previously shown in Figure 7J) due to the potential data privacy issue of an unpublished clinical trial. The Editor claimed that using unpublished clinical trial data in this study does not meet the clinical reporting standards of *Nature Communications*. In accordance with their advice, we have removed Figure 7J in the Revised Manuscript.

Thank you very much. We hope you will be satisfied with our response.

Reviewer #4 (Remarks to the Author): Expert in angiogenesis and endothelial cells

Comment 1: *The authors describe using functions within Seurat to account for batch effects between samples. Even with this processing, the clusters of thyrocytes within Figure 2 still seem to be largely described by the source tumor (Figure S2a). This would suggest that the variation within thyrocytes defining the different clusters is due to the patient rather than the cell type. This is a large problem with the analysis of the dataset that limits the strength of the conclusions on thyrocytes presented in Figures 2, 3, and 4. This means that over 75% of the transcriptional description of pre-malignant thyrocytes is provided by one sample source (and around 90% of the c01 cluster). The authors must add a detailed description of this limitation on the analysis within the text.*

Answer 1: Thanks for your comment. To avoid the possible technical noises that may affect the robustness of our study, we have carefully re-analyzed our study and we suggested that your concern regarding the patient source may be caused more by inter-tumor heterogeneity rather than the batch effect for the following reasons. In addition, **in full accordance with your advice**, we added a detailed description of this limitation on the analysis within the text of the Revised Manuscript.

First, our study performed MNN (mutual nearest neighbors) algorithm to correct the batch effect across samples, which is a commonly-used and efficient method¹. As shown in Figure 1B and Figure 1E, stromal cells and immune cells were integrated well after batch effect correction and showed little inter-sample heterogeneity, indicating that MNN has successfully removed batch effects in our study. Therefore, we suggested that the higher heterogeneity of the thyroid tumor epithelial cells (thyrocytes) between different samples was due to their biological differences, but not due to batch effects.

Second, to cover more tumor diversity, we collected samples from PTC patients with diverse clinical courses, bringing more inter-tumor heterogeneities. These

inter-tumor heterogeneities may reflect the unique biological features of samples with different clinical courses and characteristics. □ Take SC11 as example, Patient 11 had a very advanced disease and received three times of radioactive iodine (RAI), a well-known reason that induces PTC dedifferentiation. Therefore, it is easy to understand why cells from her subcutaneous metastases (SC11) sample were predominantly enriched in the c09 cluster, which had the poorest differentiation level (lowest TDS score). In other words, the predominant concentration of SC11 cells in the c09 cluster well reflects the dedifferentiated biological nature of SC11, and corresponds well with the clinical data. □ Take LN7r as another example, the right LN metastases (LN7r) of Patient 7 had prevalent and well-retained follicular architectures (Figure S6D), therefore it is easy to understand that LN7r was predominantly enriched in the c03 cluster, which had the highest differentiation level among the malignant thyrocyte clusters (c03-c09). In other words, the predominant concentration of LN7r cells in the c03 cluster well reflects the highly differentiated biological nature of LN7r, and corresponds well with the clinical data.

Third, to further clarify that the differences between these clusters reflected the inter-tumor heterogeneity or potential technical noises, we performed pathway enrichment for the c03, c05, c06 and c09 clusters, which are mainly defined by cells from a single sample. The results showed that these clusters showed distinct enriched functional pathways from the others (**Table S7**), suggesting their differences are more likely to be biological diversities of the samples.

Taken together, we think the distribution of samples across the clusters may be largely determined by the inter-tumor heterogeneity between our samples. These clinical and biological diversities between samples could provide many important findings in our study.

We totally respect your concern and admit that correcting batch effects while retaining the true biological heterogeneity is a challenge in bioinformatic analysis of single-cell sequencing. Therefore, in full accordance with your advice, we have added a detailed description of the limitations of our study in the Discussion section of the Revised Manuscript as follows.

“Second, current scRNA-seq must be performed within a short period of time after sample acquisition because fresh samples with viable cells are needed in this technology. It is still a challenge to completely correct the batch effects while fully retaining the true biological properties caused by inter-tumor heterogeneities. Though we performed the MNN algorithm for batch effect correction, batch effects may still exist in our study.”.

Thank you very much. We hope you will be satisfied with our response.

Comment 2: *How does the pseudotime statistic (and pseudotime-associated genes) describe differentiation with the clear bifurcation presented in Figure 4? It looks like lower pseudotime values correlate to follicular-like cells, how are the authors distinguishing cells with greater pseudotime values as p-EMT-like or dediff-like phenotype?*

Answer 2: Thanks for the reviewer’s question. We have provided a detailed and reasonable answer to this concern.

- 1) **Cell state is not determined by its pseudotime value:** According to the official documentation of Monocle2, it performed manifold learning to learn the trajectory that describes how cells transition from one state into another, and a cell's pseudotime value is the distance it would have to travel to get back to the root. However, the Monocle2 assumes that the trajectory has a tree structure, whose branches are not determined based on the pseudotime value, but actually the branch-specific genes.
- 2) **Distinct expression signatures between state 2 and state 3 thyrocytes:** To explore the expression signatures between state 2 and state 3 thyrocytes, we identified the differential expressed genes and pathways between these two cell types. It is found that state 2 thyrocytes showed several features of EMT, including increased expression of extracellular matrix-related genes (*SDC4*, *ECM1*, *LGALS1*), up-regulated EMT-related TFs (*HMGA2* and *EGRI*) and an enriched EMT signaling pathway. In contrast, state 3 thyrocytes had preferentially

up-regulated dedifferentiation-related TFs (*GATA2*, *MYC*, *SOX4*) and pathways (E2F TARGETS, HYPOXIA, MYC TARGETS V1).

Taken together, based on the distinct gene expression signatures (NOT pseudotime value) listed above, we defined cell state 2 and state 3 as p-EMT-like and dediff-like thyrocytes, respectively.

Thank you very much. We hope you will be satisfied with our response.

Comment 3: *For the endothelial cell phenotype assignments, it is unclear what the “immature EC” phenotype is referencing. This cluster may be describing a stalk EC phenotype, or potentially containing ECs that are proliferating. Further analysis of the differential gene expression is necessary to determine this cluster.*

Answer 3: Thanks for the reviewer’s suggestion. Your suggestion is vital to improve our understanding of EC cell phenotypes. To address your concern, we performed a literature review and also analyzed our scRNA-sequencing data. In accordance with your suggestion, both of these methods suggested that the “immature EC” resembles a stalk EC phenotype. A detailed explanation is shown as below.

First, we defined the endothelial cells in our study based on the transcription signatures from another study of lung cancer ECs ²². According to the cell type definitions in the study, the immature ECs showed a higher expression level of Notch signaling and target genes (*JAG1*, *HES1*, *ID1*, *ID2*, *ID3*), and genes involving in barrier integrity (*ENG*, *PLVAP*, *HSPG2*, *APLNR*). Therefore, they suggested that the immature cells possibly resemble the stalk-like cells, which is in full accordance with your comment.

Second, to examine whether the immature ECs are proliferating, we checked the proliferating markers in immature ECs of our study. As shown below, we found there is no higher expression of the proliferating markers (*MKI67*, *TOP2A*, *UBE2C* and *BIRC5*) in immature ECs.

In summary, we suggested that the immature ECs possibly resembled the stalk-like cells but not the proliferating ECs. In accordance with your comment, we have added the description of the immature cells in the Revised Manuscript as follows:

“while immature ECs were characterized by a higher expression level of Notch signaling and target genes (JAG1, HES1, ID1, ID2, ID3), and genes involving in barrier integrity (ENG, PLVAP, HSPG2, APLNR), which may resemble the stalk-like cells (Figure 7E).”

Thank you very much. We hope you will be satisfied with our response.

Comment 4: *The ligand/receptor relationships between tip cells and other cells are interesting. As the authors are claiming these interactions are potentially unique to tip cells, it would also be necessary to see the ligand/receptor relationships from venous and lymphatic ECs to other cell types to compare with tip cell interactions.*

Answer 4: Thanks for the reviewer’s suggestion. In full accordance with your suggestion, we have provided the VEGF-VEGFR signalings’ interactions for all five EC cell types with other cell types (**Figure S12**). As shown below, we found that tip cells showed significantly higher VEGF and VEGFR expression levels than the other EC types, suggesting that VEGF-VEGFR signalings may be unique to the tip cells.

Thank you very much. We hope you will be satisfied with our response.

tip cell

tip cell

venous

venous

immature

immature

arterial

arterial

lymphatic

lymphatic

Comment 5: *The results suggest that tip cells, that are pro-angiogenic, are also communicating with immune cells. Why would an anti-angiogenic therapy promote immune cell infiltration? Anti-angiogenic therapies should inhibit tip cell phenotypes in the tumors, effectively reducing immune crosstalk. The results on vascular-immune crosstalk should be explained in more detail, along with describing the immune cell crosstalk with other endothelial cell phenotypes. Also, validation of the effects of famitinib and camrelizumab on vascular-immune crosstalk is necessary to conclude that these drugs may be working to inhibit PTC progression by blocking these interactions. For example, in vitro analysis of sprouting angiogenesis or co-culture with endothelial cells and immune cells could be analyzed to validate this hypothesized pathway.*

Answer 5: Thanks for the reviewer's comments. This is an interesting and constructive concern. To address your concern, we performed a literature review and also analyzed our scRNA-sequencing data. A detailed explanation was shown as below.

First, tip cells may inhibit immune cells adhesion and migration. Recent studies have indicated that intra-tumor vessels (which are made of tip cells) are irregular and leaky, inducing multiple immune suppression factors, such as TGF- β , VEGF, leading to the developmental deficiency of vascular walls and the inhibition of immune cells adhesion and migration²³. Therefore, if treated with anti-angiogenesis treatment, the tip cells would be inhibited and the tumor vessels could be normalized, increasing the tumor-infiltrating lymphocytes as previous studies suggested^{24, 25}. As a result, we suggested that the increased immune cell infiltration after anti-angiogenic therapy is not contradictory to the crosstalk between tip cells and immune cells.

Second, in full accordance with your suggestion, we have added a detailed description of the crosstalks between other ECs and immune cells in the revised manuscript as follows.

In this analysis, we observed widespread interactions between ECs and immune cells. For instance, we found that the lymphatic ECs interacted with immune cells

through the atypical chemokine receptor 2 (ACKR2), which has been reported to regulate chemokine availability (Figure S10). Meanwhile, we found that venous, immature and arterial ECs interacted with immune cells through the Intercellular Adhesion Molecule 1 (ICAM1) on its surface, while the ICAM1 expression was significantly reduced in tip cells and lymphatic ECs (Figure S11). In particular, we identified that the crosstalks between tip ECs and immune cells were predominantly achieved through the key angiogenic VEGF-VEGFR signalings (Figure 7I and Figure S12).

Finally, as suggested by the Editor and other reviewers, we have removed our *in vivo* experiments in the Revised Manuscript. We admitted that the lack of *in vivo* and *in vitro* validation is one of the limitations of our study, and we have added a detailed description of the limitations in the revised manuscript as follows.

*“At last, our study performed a series of bioinformatic tools to dissect the TME of PTC, while the functional validation of these findings is lacking, such as the iodine uptake and retention in different kinds of thyrocytes, and the key genes in PTC tumorigenesis. Further *in vivo* and *in vitro* experiments are awaited to validate these findings in the future.”*

Thank you very much. We hope you will be satisfied with our response.

Comment 6: *The authors highlight TMSB4X as a potential marker for malignancy by showing expression levels in c01, c02, and c03-c09 in Figure 3c. But this analysis does not consider single cells as they are transitioning. A more accurate visualization would be to show TMSB4X expression in single cells graphed on Component 1 versus Component 2, as shown in Figure 3f. Additionally, is there differential expression of TMSB4X within the malignant cell clusters shown in Figure 4?*

Answer 6: Thanks for the reviewer’s suggestion. In full accordance with your comment, we have added the expression levels of TMSB4X for each thyrocyte graphed on Component 1 vs. Component 2 based on the trajectory analysis in **Figure S5I** of the Revised Manuscript, as shown below.

Meanwhile, in full accordance with your comment, we explored the differential expression of *TMSB4X* in the State 1-3 malignant cell clusters, and found the expression level of *TMSB4X* gradually increased from State 1 to State 2 to State 3, suggesting the potential role of *TMSB4X* in PTC progression (as shown below). We have added this result as **Figure S5J** and added a description in the Revised Manuscript as follows.

“Moreover, TMSB4X, a potential biomarker of PTC progression as described above, was also expressed at the highest level in State 3 (Figure S5I and S5J).”

Thank you very much. We hope you will be satisfied with our response.

Comment 7: *Although the Methods section suggests that tSNE and UMAP dimensionality reduction analyses were performed when appropriate, it is not clear from the text why t-SNE was shown in Figures 1, 3, 6, and 7, but UMAP was shown in Figure 2.*

Answer 7: Thanks for the reviewer’s comment. We will provide a detailed explanation.

First, both tSNE and UMAP are good and commonly-used dimensionality reduction and visualization methods in scRNA-seq studies. However, an advantage of UMAP is that it can preserve a better global structure than tSNE²⁶.

Second, in Figure 2, we would like to preserve a better global structure to reveal the distinctions between normal and tumor cells. In other words, the differences

between c01-c02 and the remaining clusters could be directly visualized by the distances in the UMAP image. That is why we performed the UMAP visualization rather than tSNE in Figure 2.

Third, in accordance with your comment, we have also performed the tSNE visualization for all the thyrocytes as below. As the figure shown below, both UMAP and tSNE could reveal the differences between these nine clusters.

Thank you very much. We hope you will be satisfied with our response.

Comment 8: *With the color choices, the subcutaneous population is difficult to see in Figure 1e (right panel).*

Answer 8: Thanks for the reviewer's suggestion. In full accordance with your suggestion, we have changed the color choices in Figure 1E (right panel) in the Revised Manuscript.

Thank you very much. We hope you will be satisfied with our response.

Comment 9: *More description of canonical markers used to identify different clusters as phenotypic populations would be helpful in all experiments.*

Answer 9: Thanks for the reviewer's suggestion. In full accordance with your

suggestion, we have provided more description of canonical markers to identify different cell clusters in the Revised Manuscript as follows.

“After integrating the transcriptional data from all acquired cells, we primarily applied low-resolution t-distributed stochastic neighbor embedding (t-SNE) clustering and identified six main cell populations, which were labeled as T/natural killer (NK) cells (CD3D, CD3E, CD3G, CD247), B cells (CD79A, CD79B, IGHM, IGHD), thyrocytes (TG, EPCAM, KRT18, KRT19), myeloid cells (LYZ, S100A8, S100A9, CD14), fibroblasts (COL1A1, COL1A2, COL3A1, ACTA2) and endothelial cells (PECAM1, CD34, CDH5, VWF) (Figures 1C, 1D and S1C).”

Thank you very much. We hope you will be satisfied with our response.

Reviewer #5 (Remarks to the Author): Expert in single-cell RNA-seq and microenvironment analysis

Comment 1: *Trajectory analysis: a. An important aspect to note is that any trajectory detection algorithm assumes that cells lie along a continuous manifold and when start cell is specified directionality is known. However, the authors use the application of trajectory detection as proof of existence of a continuous manifold - I think the authors should correct the interpretation here and demonstrate the existence of continuous manifolds using tools such as gene expression trends of known markers.*

Answer 1: Thanks for the reviewer’s suggestion. In full accordance with your suggestion, we have corrected the interpretation of the trajectory analysis in the Revised Manuscript as follows:

“Moreover, trajectory analysis further suggested the potential transitions between c01 and c02 population (Figure 3B).”

Furthermore, in full accordance with your comment, we have shown the gene expression trends of known markers (TDS-associated genes and reported PTC tumor markers) in **Figure S5F** of the Revised Manuscript. Based on the new **Figure 5F** below, we suggested that the thyrocytes follow a continuous manifold in our study.

Thank you very much. We hope you will be satisfied with our response.

Comment 2: *b. Are results presented in Figs. 3e-f reproducible across different patient samples?*

Answer 2: Thanks for the reviewer's question. We are sorry because the results presented in Figs. 3e-f may not be reproduced using other samples in the present study due to two following reasons.

First and most important, results of Figs. 3E-F showed premalignant thyrocytes that represented an important finding of our study. These premalignant cells can only be identified when the following three criteria were simultaneously met. (1) histologically normal follicular architecture (outwardly normal); (2) different from c01 (normal thyrocytes) in the evolutionary paths; (3) the most important, the potency of multifocal independent tumorigenesis confirmed by pathologists. Samples simultaneously meeting these criteria is rare in clinical practice. In our study, only Patient 5 met all these criteria.

Second, due to the difficulties in sample collection, we obtained the paratumor, tumor and metastatic lymph node samples simultaneously only from Patient 5. Therefore, we could not present similar results in Figures 3E-3F for other samples.

In summary, for the two points above, we must admit that the results in Figure 3E-3F need to be further verified using more samples in our future studies. Therefore, in full accordance with your comment, we have acknowledged the weakness in the Revised Manuscript as follows.

"We acknowledge that our study has several limitations. First, the sample size and cell number in our study is limited, and further expansion of the study cohort is required to better elucidate the tumor progression and diversity in PTC. In particular, the premalignant cells found in our study are primarily determined by one patient, and this state of thyrocytes should be verified in further studies."

Thank you very much. We hope you will be satisfied with our response.

Comment 3: *The authors switch from Monocle to Diffusion maps for analysis of*

paranormal thyrocytes to all thyrocytes - why was this switch necessary and why not use the same analysis approaches across the board?

Answer 3: Thanks for the reviewer's correction. In our study, we performed the trajectory analysis using Monocle2 all the time and did not perform the diffusion map method. We are sorry for the mistakes in the manuscript and we have changed it in the Revised Manuscript.

Thank you very much. We hope you will be satisfied with our response.

Comment 4: *I am also surprised to see that the cluster structure is rather disrupted in Fig. 4b - in particularly, c08 appears to be present in both branches. This indicates either the clustering or trajectory detection needs some correction since trajectory detection in general does not dramatically alter the relationships between cells.*

Answer 4: Thanks for the reviewer's correction. In full accordance with your suggestion, we have summarized the thyrocyte distribution in three states as in **Table S13** in the Revised Manuscript.

According to the table:

- 1) c01 and c02 thyrocytes are almost in State 1;
- 2) c03 thyrocytes were mainly distributed into State 1 and State 2;
- 3) c04 thyrocytes were distributed to all three states;
- 4) c05 and c08 thyrocytes mainly belonged to State 2;
- 5) the majority of c06, c07 and c09 thyrocytes belong to State 3.

Based on the table and figure showing the thyrocyte distribution below, we found that the majority of the c08 cells are distributed to State 2, and the trajectory detection did not dramatically alter the relationships between cells. Therefore, we suggested that the clustering and trajectory analysis in our study are consistent.

Thank you very much. We hope you will be satisfied with our response.

Table. The distribution of cell clusters in different cell states

State	c01	c02	c03	c04	c05	c06	c07	c08	c09
State 1	4649	1317	593	982	31	284	7	9	1
State 2	0	1	868	2331	3148	114	76	5741	75
State 3	3	16	36	982	126	3407	6634	253	4581

Comment 5: *Batch correction: In the first step, the authors apply batch correction (Fig. 1) - which inherently assumes all differences are technical. However, when zooming into thyrocytes the authors do not apply any batch correction (Fig. 2). The authors should clarify the assumptions and approach here - perhaps they could perform differential analysis between the thyrocyte patient samples and use gene set enrichment or similar approaches to examine whether differences are technical or if they represent true biology.*

Answer 5: Thanks for your comment. To avoid the possible technical noises that may affect the robustness of our study, we have carefully re-analyzed our study and we

suggested that the patient source may be caused more by inter-tumor heterogeneity rather than the batch effect for the following reasons.

First, our study performed MNN (mutual nearest neighbors) algorithm to correct the batch effect across samples, which is a commonly-used and efficient method ¹. As shown in Figure 1B and Figure 1E, stromal cells and immune cells were integrated well after batch effect correction and showed little inter-sample heterogeneity, indicating that MNN has successfully removed batch effects in our study. Therefore, we suggested that the higher heterogeneity of the thyroid tumor epithelial cells (thyrocytes) between different samples was due to their biological differences, but not due to batch effects.

Second, to cover more tumor diversity, we collected samples from PTC patients with diverse clinical courses, bringing more inter-tumor heterogeneities. These inter-tumor heterogeneities may reflect the unique biological features of samples with different clinical courses and characteristics. □ Take SC11 as example, Patient 11 had a very advanced disease and received three times of radioactive iodine (RAI), a well-known reason that induces PTC dedifferentiation. Therefore, it is easy to understand why cells from her subcutaneous metastases (SC11) sample were predominantly enriched in the c09 cluster, which had the poorest differentiation level (lowest TDS score). In other words, the predominant concentration of SC11 cells in the c09 cluster well reflects the dedifferentiated biological nature of SC11, and corresponds well with the clinical data. □ Take LN7r as another example, the right LN metastases (LN7r) of Patient 7 had prevalent and well-retained follicular architectures (Figure S6D), therefore it is easy to understand that LN7r was predominantly enriched in the c03 cluster, which had the highest differentiation level among the malignant thyrocyte clusters (c03-c09). In other words, the predominant concentration of LN7r cells in the c03 cluster well reflects the highly differentiated biological nature of LN7r, and corresponds well with the clinical data.

Third, in full accordance with your suggestion, we performed pathway enrichment analysis for the c03, c05, c06 and c09 clusters, which are mainly defined by cells from a single sample, to examine whether differences between these clusters

are technical or representing true biology as follows (**Table S7**).

c03: adipogenesis, oxidative phosphorylation, and xenobiotic metabolism are up-regulated, while the TNF α signaling via NF κ B, apoptosis, hypoxia, mTORc1 signaling, and P53 pathway were down-regulated.

c05: Coagulation, apoptosis pathways were up-regulated, and oxidative phosphorylation, TGF beta signaling were down-regulated.

c06: MYC targets v1 and mTORc1 signaling pathways were up-regulated, and the epithelial-mesenchymal transition, reactive oxygen species pathways were down-regulated.

c09: MYC targets v1, oxidative phosphorylation, P53 pathway, TNF α signaling via NF κ B, apoptosis, and reactive oxygen species pathway were significantly up-regulated, while the protein secretion and interferon-alpha response were down-regulated.

According to the pathway enrichment analysis results, we found that these clusters showed distinct enriched functional pathways from the others, suggesting their differences are not technical but more likely to be true biological significance. In the Revised Manuscript, we have added the description as follows.

“Moreover, we found that c03, c05, c06 and c09 are mainly defined by cells from a single sample. Therefore, we then performed the pathway enrichment analysis using the DEGs for each cluster and identified distinct up-regulated and down-regulated pathways for each cluster, suggesting that their differences were more attributed to biological diversities cause by inter-tumor heterogeneities rather than batch effects (Tables S6 and S7).”

Fourth, MNN does not encourage the users to subset the dataset and re-correct because it may eliminate genuine differences between subclusters (<https://github.com/MarioniLab/FurtherMNN2018/issues/6>). Therefore, to better preserve the biological significance, we did not perform the batch effect correction method again after zooming into thyrocytes.

Taken together, we think the distribution of samples across the clusters may be largely determined by the inter-tumor heterogeneity between our samples. These

clinical and biological diversities between samples could provide many important findings in our study.

We totally respect your concern and admit that correcting batch effects while retaining the true biological heterogeneity is a challenge in bioinformatic analysis of single-cell sequencing. Therefore, in full accordance with your advice, we have added a detailed description of the limitations of our study in the Discussion section of the Revised Manuscript as follows.

“Second, current scRNA-seq must be performed within a short period of time after sample acquisition because fresh samples with viable cells are needed in this technology. It is still a challenge to completely correct the batch effects while fully retaining the true biological properties caused by inter-tumor heterogeneities. Though we performed the MNN algorithm for batch effect correction, batch effects may still exist in our study.”

Thank you very much. We hope you will be satisfied with our response.

Comment 6: *In a couple of instances, the authors choose to highlight genes as being relevant to the biology of disease: (a) TMSB4X in Fig2; (b) PKHD1L1 in Fig. 3. What was the statistical criteria for these genes and are there genes at a similar level of significance? The authors better need to justify the selection of these genes given the weight of biological role attributed to them.*

Answer 6: Thanks for the reviewer’s suggestion. In full accordance with your suggestion, we will explain our statistical criteria in detail as below.

1) **TMSB4X:** In Figure 2B, we would like to identify the upregulated genes in malignant thyrocytes. The criteria for identifying the differential expressed genes are as follows: \log_2 fold-change > 2 and FDR < 0.05. In total, we identified eight genes, including (*S100A4*, *FN1*, *TMSB4X*, *APOE*, *CXCL14*, *TIMP1*, *LGALS3*, *IGFBP6*). Among the eight genes, all other seven genes have been reported in PTC tumorigenesis, while only the role of TMSB4X in PTC is unclear. Therefore, we highlighted the *TMSB4X* in our study for further analysis. In full accordance

with your suggestion, we added the differential expressed genes between tumor and normal thyrocytes as in **Table S11** in the revised version. Moreover, the criteria were supplemented in the Revised Manuscript as follows (red texts are the supplemented part):

“Among the most dysregulated DEGs (\log_2 fold-change > 2, FDR < 0.05) between different origins of thyrocytes, we found TMSB4X, which has not been reported in PTC, was significantly upregulated in malignant cells than in non-malignant cells (Figure 2F and Table S11).”

2) **PKHD1L1**: To identify the marker genes for the premalignant thyrocytes, we set the criteria for differential expressed genes as follows: $|\log_2$ fold-change > 1 and FDR < 0.05. In total, we have identified 20 DEGs. However, we found that only 3.5% of the premalignant cells expressed **PKHD1L1**, suggesting that **PKHD1L1** is significantly decreased in the premalignant cells. Therefore, we further concentrated on **PKHD1L1** and its role in PTC. In full accordance with your suggestion, we have added the differential expressed genes between premalignant and normal thyrocytes as **Table S12** in the Revised Manuscript. Meanwhile, the criteria were supplemented in the Revised Manuscript as follows (red texts are the supplemented part):

“The most dysregulated genes ($|\log_2$ fold-change| > 1, FDR < 0.05) between normal (c01) and premalignant (c02) thyrocytes reflected the underlying early-onset transcriptional changes during PTC tumorigenesis, in which PKHD1L1 was significantly downregulated in premalignant and malignant thyrocytes (Figures S4C and Table S12).”

Thank you very much. We hope you will be satisfied with our response.

Comment 7: *Classification of scRNA-seq data using bulk data: While the bulk data classifier really well, how does the authors take into account the drop-outs in scRNA-seq when transferring the model? It would be useful to examine the prediction scores to ensure dropouts do not adversely affect the single-cell annotation.*

Answer 7: Thanks for the reviewer's question. We totally agree with your comment that the sparseness of single-cell sequencing data may adversely affect prediction accuracy. In our study, we have taken full account of it and performed several methods to overcome the adverse effects induced by the sparseness of the scRNA-sequencing dataset as much as possible. A detailed explanation was shown below.

- 1) **First**, we picked the top 2000 variable genes in the scRNA-seq data, which decreased the dropout percentages of the dataset, and utilized these genes for model training to discriminate tumor and normal samples from bulk RNA-seq data.
- 2) **Second**, we utilized the Spearman correlation as a measurement, which focused on ranks but not the linear relationships between two variables, to assess the correlation between single-cell expression and bulk RNA-seq data. Similarly, multiple single-cell annotation methods, such as scMatch and SingleR, also performed the Spearman correlation analysis to annotate cells^{27, 28}.
- 3) **Finally**, we performed the KNN (K nearest neighbors) method to further improve the robustness of our method. We annotated each cell according to their nearest neighbors but not the prediction scores, which reduced the adverse effects of the drop-outs in scRNA-seq data.

In this way, we suggested we could, to some extent, avoid the adverse effects of the drop-outs in scRNA-seq in our model.

Thank you very much. We hope you will be satisfied with our response.

References in this Response Letter

1. Haghverdi L, Lun ATL, Morgan MD, Marioni JC. Batch effects in single-cell RNA-sequencing data are corrected by matching mutual nearest neighbors. *Nat Biotechnol* **36**, 421-427 (2018).
2. Perng YC, Lenschow DJ. ISG15 in antiviral immunity and beyond. *Nature reviews Microbiology* **16**, 423-439 (2018).
3. D'Cunha J, Knight E, Jr., Haas AL, Truitt RL, Borden EC. Immunoregulatory properties of ISG15, an interferon-induced cytokine. *Proceedings of the National Academy of Sciences of the United States of America* **93**, 211-215 (1996).
4. Padovan E, *et al.* Interferon stimulated gene 15 constitutively produced by melanoma cells induces e-cadherin expression on human dendritic cells. *Cancer research* **62**, 3453-3458 (2002).
5. Wagle MC, *et al.* A transcriptional MAPK Pathway Activity Score (MPAS) is a clinically relevant biomarker in multiple cancer types. *NPJ Precis Oncol* **2**, 7 (2018).
6. Ping Q, *et al.* Cancer-associated fibroblasts: overview, progress, challenges, and directions. *Cancer Gene Ther*, (2021).

7. Han C, Liu T, Yin R. Biomarkers for cancer-associated fibroblasts. *Biomark Res* **8**, 64 (2020).
8. Szabo PA, *et al.* Single-cell transcriptomics of human T cells reveals tissue and activation signatures in health and disease. *Nat Commun* **10**, 4706 (2019).
9. Schafflick D, *et al.* Integrated single cell analysis of blood and cerebrospinal fluid leukocytes in multiple sclerosis. *Nat Commun* **11**, 247 (2020).
10. Lin W, *et al.* Single-cell transcriptome analysis of tumor and stromal compartments of pancreatic ductal adenocarcinoma primary tumors and metastatic lesions. *Genome Med* **12**, 80 (2020).
11. Baran Y, *et al.* MetaCell: analysis of single-cell RNA-seq data using K-nn graph partitions. *Genome Biol* **20**, 206 (2019).
12. Muhl L, *et al.* Single-cell analysis uncovers fibroblast heterogeneity and criteria for fibroblast and mural cell identification and discrimination. *Nat Commun* **11**, 3953 (2020).
13. Helle E, Ampuja M, Antola L, Kivela R. Flow-Induced Transcriptomic Remodeling of Endothelial Cells Derived From Human Induced Pluripotent Stem Cells. *Front Physiol*

11, 591450 (2020).

14. Wang Z, *et al.* Single-cell RNA sequencing reveals a novel cell type and immunotherapeutic targets in papillary thyroid cancer. *medRxiv*, 2021.2002.2024.21251881 (2021).
15. Peng M, *et al.* Single-cell transcriptomic landscape reveals the differences in cell differentiation and immune microenvironment of papillary thyroid carcinoma between genders. *Cell & bioscience* **11**, 39 (2021).
16. Dunn LA, *et al.* Vemurafenib Redifferentiation of BRAF Mutant, RAI-Refractory Thyroid Cancers. *J Clin Endocrinol Metab* **104**, 1417-1428 (2019).
17. Montero-Conde C, *et al.* Relief of feedback inhibition of HER3 transcription by RAF and MEK inhibitors attenuates their antitumor effects in BRAF-mutant thyroid carcinomas. *Cancer Discov* **3**, 520-533 (2013).
18. Hossain MA, *et al.* Network-Based Genetic Profiling Reveals Cellular Pathway Differences Between Follicular Thyroid Carcinoma and Follicular Thyroid Adenoma. *Int J Environ Res Public Health* **17**, (2020).
19. Opitz R, Maquet E, Zoenen M, Dadhich R, Costagliola S. TSH receptor function is

required for normal thyroid differentiation in zebrafish. *Mol Endocrinol* **25**, 1579-1599 (2011).

20. Ishii J, *et al.* Congenital goitrous hypothyroidism is caused by dysfunction of the iodide transporter SLC26A7. *Commun Biol* **2**, 270 (2019).
21. Zheng C, Quan R, Xia EJ, Bhandari A, Zhang X. Original tumour suppressor gene polycystic kidney and hepatic disease 1-like 1 is associated with thyroid cancer cell progression. *Oncol Lett* **18**, 3227-3235 (2019).
22. Goveia J, *et al.* An Integrated Gene Expression Landscape Profiling Approach to Identify Lung Tumor Endothelial Cell Heterogeneity and Angiogenic Candidates. *Cancer Cell* **37**, 21-36 e13 (2020).
23. Munn LL, Jain RK. Vascular regulation of antitumor immunity. *Science* **365**, 544-545 (2019).
24. Huang Y, *et al.* Vascular normalizing doses of antiangiogenic treatment reprogram the immunosuppressive tumor microenvironment and enhance immunotherapy. *Proc Natl Acad Sci U S A* **109**, 17561-17566 (2012).
25. Shigeta K, *et al.* Dual Programmed Death Receptor-1 and Vascular Endothelial

Growth Factor Receptor-2 Blockade Promotes Vascular Normalization and Enhances Antitumor Immune Responses in Hepatocellular Carcinoma. *Hepatology* **71**, 1247-1261 (2020).

26. Becht E, *et al.* Dimensionality reduction for visualizing single-cell data using UMAP. *Nat Biotechnol*, (2018).
27. Aran D, *et al.* Reference-based analysis of lung single-cell sequencing reveals a transitional profibrotic macrophage. *Nat Immunol* **20**, 163-172 (2019).
28. Hou R, Denisenko E, Forrest ARR. scMatch: a single-cell gene expression profile annotation tool using reference datasets. *Bioinformatics* **35**, 4688-4695 (2019).

REVIEWERS' COMMENTS

Reviewer #2 (Remarks to the Author):

All my concerns were satisfactorily addressed and I appreciate the authors for the necessary changes made to the manuscript.

Reviewer #3 (Remarks to the Author):

The mole of work done from the authors is highly appreciated. However, for a scientific rigor stand of view there are still limitations. Overall, this is a descriptive study without new mechanistic insights. The translational applications of the descriptive results are not clear.

As also reported from the authors the study has a central limitation due the low number of patients enrolled in the design.

Importantly, there is also a lack of a thyroid-specific mouse model to validate and strength the results. The new genes added as TDF (thyroid differentiation factors) unfortunately were not validated through the major assays, i.e. iodide uptake at least in vitro and ATC re-differentiation analysis.

Some statistical analysis is still weak and not straightforward in supporting the biology of the models.

Reviewer #4 (Remarks to the Author):

The authors have sufficiently addressed the concerns raised, and the changes to the text and figures help to clarify the data and interpretations thereof.

Reviewer #5 (Remarks to the Author):

The authors have satisfactorily address the concerns raised in my review. There are no additional concerns about the analysis approaches and interpretation of the results presented in the manuscript. I cannot speak about the novelty of the biological findings since this is outside my area of expertise.

Responses to the Reviewers

Reviewer #2 (Remarks to the Author): Expert in thyroid cancer genetics, genomics, and therapy

Comment 1: *All my concerns were satisfactorily addressed and I appreciate the authors for the necessary changes made to the manuscript.*

Answer 1: Thanks very much for your constructive suggestions, which significantly improved the quality of our manuscript. We are happy that you are satisfied with our response.

Reviewer #3 (Remarks to the Author): Expert in thyroid cancer genetics, therapy, mouse models, and angiogenesis

Comment 1: *The mole of work done from the authors is highly appreciated. However, for a scientific rigor stand of view there are still limitations. Overall, this is a descriptive study without new mechanistic insights. The translational applications of the descriptive results are not clear.*

Answer 1: Thanks for your question. Our study profiled transcriptomes of 158,577 cells from 11 patients' paratumors, localized/advanced tumors, initially-treated/recurrent lymph nodes, and radioactive iodine (RAI)-refractory distant metastases, covering comprehensive clinical courses of PTC. Using this resource, we have suggested that the use of immunotherapy and its combination with anti-angiogenic therapy in PTC might be promising, which provides new mechanistic insights and potential translational applications. In the future, we would perform *in vivo* and *in vitro* studies and clinical trials to further confirm our findings.

Thank you very much. We hope you will be satisfied with our response.

Comment 2: *As also reported from the authors the study has a central limitation due to the low number of patients enrolled in the design.*

Answer 2: Thanks for your question. In this study, we profiled a total of 158,577 high-quality cells. We suggest that the number of cells might be sufficient to identify the PTC diversity to some extent. We agree with your suggestion that the sample size is limited in our study to answer important biological questions (tumor diversity and progression). In the future, we would perform the scRNA-sequencing study of PTC using a larger sample size to verify our findings further and identify more PTC diversities. Therefore, we have added the description of our limitations in the manuscript as follows:

“We acknowledge that our study has several limitations. First, the sample size and

cell number in our study is limited, and further expansion of the study cohort is required to better elucidate the tumor progression and diversity in PTC. ”.

Thank you very much. We hope you will be satisfied with our response.

Comment 3: *Importantly, there is also a lack of a thyroid-specific mouse model to validate and strength the results.*

Answer 3: Thanks for your suggestion. We agree with your suggestion that we should validate and strengthen our results using a thyroid-specific mouse model. We will perform these *in vivo* studies in the near future to validate our conclusions further. Therefore, we have added the lack of mice model as a limitation in our manuscript as follows.

“At last, our study performed a series of bioinformatic tools to dissect the TME of PTC, while the functional validation of these findings is lacking, such as the iodine uptake and retention in different kinds of thyrocytes, and the key genes in PTC tumorigenesis. Further in vivo and in vitro experiments are awaited to validate these findings in the future.”.

Thank you very much. We hope you will be satisfied with our response.

Comment 4: *The new genes added as TDF (thyroid differentiation factors) unfortunately were not validated through the major assays, i.e. iodide uptake at least in vitro and ATC re-differentiation analysis.*

Answer 4: Thanks for your suggestion. We agree with your suggestion that these newly identified TDS-associated genes should be further verified through *in vitro* and *in vivo* studies. We will try to verify the roles of these genes in thyroid differentiation in our future studies. Therefore, we have added it as a limitation in our revised manuscript as follows.

“At last, our study performed a series of bioinformatic tools to dissect the TME of PTC, while the functional validation of these findings is lacking, such as the iodine

uptake and retention in different kinds of thyrocytes and the roles of the newly identified TDS-associated genes in PTC tumorigenesis. ”.

Thank you very much. We hope you will be satisfied with our response.

Comment 5: *Some statistical analysis is still weak and not straightforward in supporting the biology of the models.*

Answer 5: Thanks for your suggestion. In our study, we performed stringent quality control procedures and applied both doublet detection and batch effect correction methods to remove the unwanted variations, to ensure that our results and conclusions would not be biased. In summary, we feel that our analyzing pipelines are appropriate and reliable. In the future, we would perform the scRNA-sequencing study of PTC using larger sample size, and verify our findings using both *in vivo* and *in vitro* models further.

Thank you very much. We hope you will be satisfied with our response.

Reviewer #4 (Remarks to the Author): Expert in angiogenesis and endothelial cells

Comment 1: *The authors have sufficiently addressed the concerns raised, and the changes to the text and figures help to clarify the data and interpretations thereof.*

Answer 1: Thanks very much for your constructive suggestions, which significantly improved the quality of our manuscript. We are happy that you are satisfied with our response.

Reviewer #5 (Remarks to the Author): Expert in single-cell RNA-seq and microenvironment analysis

Comment 1: *The authors have satisfactorily addressed the concerns raised in my review. There are no additional concerns about the analysis approaches and interpretation of the results presented in the manuscript. I cannot speak about the novelty of the biological findings since this is outside my area of expertise.*

Answer 1: Thanks very much for your constructive suggestions, which significantly improved the quality of our manuscript. We are happy that you are satisfied with our response.